# Detection of Saharan dust and biomass burning events using near real-time intensive aerosol optical properties in the northwestern Mediterranean.

M. Ealo[1 2], A. Alastuey[1], A. Ripoll[1], N. Pérez[1], M.C. Minguillón[1], X. Querol[1] and M. Pandolfi[1].

[1] Institute of Environmental Assessment and Water Research (IDAEA-CSIC), Barcelona, Spain
[2] Department of Astronomy and Meteorology, Faculty of Physics, University of Barcelona, Spain

Correspondence *to*: M. Ealo (marina.ealo@idaea.csic.es)

**Abstract.** The study of Saharan dust events (SDE) and biomass burning (BB) emissions are both topic of great scientific interest since they are frequent and important polluting scenarios affecting air quality and climate. The main aim of this work is evaluating the feasibility of using near real-time in situ aerosol optical measurements for the detection of these atmospheric events in the Western Mediterranean Basin (WMB). With this aim, intensive aerosol optical properties (SAE: scattering Ångström exponent, AAE: absorption Ångström exponent, SSAAE: single scattering albedo Ångström exponent, and g: asymmetry parameter) were derived from multi-wavelength aerosol light scattering, hemispheric backscattering and absorption measurements performed at regional (Montseny; MSY, 720 m a.s.l.) and continental (Montsec; MSA, 1570 m a.s.l.) background sites in the WMB. A sensitivity study aiming at calibrating the measured intensive optical properties for SDE and BB detection is presented and discussed.

The detection of Saharan dust events (SDE) by means of the SSAAE parameter and Ångström matrix depended on the altitude of the measurement station, and on SDE intensity. At MSA (mountain-top site) SSAAE detected around 85% of SDE compared with 50% at MSY station, where pollution episodes dominated by fine anthropogenic particles frequently masked the effect of mineral dust on optical properties during less intense SDE. Furthermore, an interesting feature of SSAAE was its capability to detect the presence of mineral dust after the end of SDE. Thus, resuspension processes driven by summer regional atmospheric circulations and dry conditions after SDE favored the accumulation of mineral dust at regional level having important consequences for air quality. On average, SAE, AAE and *g* ranged between -0.7 and 1, 1.3 and 2.5, and 0.5 and 0.75, respectively, during SDE.

Based on the Aethalometer model, biomass burning (BB) contribution to equivalent black carbon (BC) accounted for 36% and 40% at MSY and MSA respectively. Linear relationships were found between AAE and %$BC_{bb}$, with AAE values reaching around 1.5 when %$BC_{bb}$ was higher than 50%. BB contribution to organic matter (OM) at MSY was around 30%. Thus FF combustion sources showed important contributions to both BC and OM in the region under study. Results for OM source apportionment showed good agreement with simultaneous biomass burning organic aerosol (BBOA) and hydrocarbon-like organic aerosol (HOA) calculated from Positive Matrix Factorization (PMF) applied to simultaneous

Aerosol Mass Spectrometer (ACSM) measurements. A wildfire episode was identified at MSY, showing AAE values up to 2 when daily BB contributions to BC and OM were 73% and 78% respectively.

## 1. Introduction

Atmospheric aerosols play an important role in our environment affecting air quality and health (Pope and Dockery, 2006), contributing to the largest uncertainties to the total radiative forcing (IPCC 2007, 2013). Aerosol affects climate by perturbation on the Earth's radiative budget, directly through absorption and scattering of solar and terrestrial radiation, and indirectly by acting as cloud condensation nuclei (Twomey et al., 1984; Albrecht, 1989). Most particles scatter the sunlight, causing a net cooling at the top of the atmosphere (TOA), whereas black carbon (BC) absorbs solar radiation in the whole visible spectrum, thus causing a net warming at the TOA (Jacobson, 2001; Ramanathan and Carmichael, 2008; Bond et al., 2013). Absorbing particles can modify the radiation fluxes directly, by absorption of shortwave solar radiation and semi-directly, by modifying the temperature distribution of the atmosphere. Absorption in the UV range is important, since it may affect photo-chemistry thus reducing tropospheric ozone concentration (Jacobson, 1998; Chen and Bond, 2010). Mineral matter and some organic compounds mainly from biomass burning (BB) emissions, called brown carbon (BrC), can also absorb solar radiation in the UV range of the solar spectrum. BrC contains a large and variable group of organic compounds including humic substances, polyaromatic hydrocarbons, and lignin (Andreae and Gelencsér, 2006), and it is formed by inefficient combustion of hydrocarbons (biomass burning) and also by photo-oxidation of biogenic particles (Yang et al., 2009). The light absorption by mineral dust depends on its content of ferric oxides (Sokolik and Toon, 1999; Alfaro et al., 2004).

Thus, the study of the relationship between physicochemical and optical properties of aerosols is strongly required in order to obtain a deeper characterization of atmospheric aerosols and therefore a better estimation of their radiative forcing. Some parameters can be derived from multi-wavelength scattering and absorption aerosol measurements in order to describe the optical properties as a function of the wavelength. These parameters, such as single scattering albedo (SSA), asymmetry parameter (g), scattering Ångström exponent (SAE), absorption Ångström exponent (AAE) and single scattering albedo Ångström exponent (SSAAE), are determined by the physicochemical properties of aerosols and are called intensive because do not depend on the particle mass. As a consequence the intensive aerosol optical properties are a valuable input for climate models which require accurate information concerning the variability of atmospheric composition for targeted species via comparison with observations (Laj et al., 2009). Given the huge variety of aerosol emission sources and formation and transformation processes, there is a substantial need of accurate real-time aerosol optical measurements to achieve a low-error estimation of the effects that atmospheric particles have on climate coupling experimental measurements and modelling results (IPCC, 2007; 2013).

In order to get a wide coverage of the spatial variability of aerosols, aerosol optical data are obtained all over the world from both in situ and remote measurements. The in situ optical measurements are usually performed in international networks by

means of automatic instruments which provide real-time data at high temporal and spatial resolution. Some of the most relevant networks are Aerosols, Clouds and Trace Gases Research InfraStructure (ACTRIS; www.actris.net), Global Atmospheric Watch (GAW; www.gaw-wdca.org), Aerosol Robotic Network (AERONET; www.aeronet.gsfc.nasa.gov) and NOAA baseline observatory (www.esrl.noaa.gov).

The WMB is affected by a large variety of emissions sources: natural sources such as Saharan dust, marine aerosols and wildfire; industrial and urban emissions from densely populated areas in the coastline; and transboundary from the European continent (Steinbrecher et al., 2009; Rodríguez et al., 2011; Pey et al., 2013a , 2013b; Garcia-Hurtado et al., 2014). The atmospheric dynamics coupled to local orography gives rise to a complex mixture of pollutants (Millán et al., 1997) where aerosol formation and transformation processes take place and accumulation of pollutant is very frequent (Rodríguez et al., 2002, 2003; Pérez et al., 2004; Jiménez et al., 2006; Pey et al., 2010; Jorba et al., 2013; Pandolfi et al., 2014b). Furthermore, the high occurrence of Saharan dust events (SDE), especially during the summer period, also contribute strongly to the increment of PM10 levels in the WMB (Rodríguez et al., 2001, 2015; Querol et al., 2009; Pey et al., 2013a). In fact, more than 70 % of the exceedances of the PM10 daily limit value (2008/50/CE European Directive) at most regional background sites of Spain have been attributed to dust outbreaks (Escudero et al., 2007a). Thus, all these processes lead to a radiative forcing in the WMB among the highest in the word (Jacobson, 2001). Nevertheless there is a large uncertainty in the total radiative forcing by atmospheric aerosols in the Mediterranean area (Mallet et al., 2013). The high occurrence and intensity of SDE in the WMB give us the opportunity to look deeply into the characterization of the optical properties of mineral dust when mixed with local aerosols. Despite several studies published on physical and chemical properties of mineral dust in the WMB region (Rodríguez et al., 2001; Escudero et al., 2007b; Querol et al., 2009; Pey et al., 2013a), very few have studied how SDE affect the aerosol intensive optical properties (Pandolfi et al., 2011 and 2014a; Valenzuela et al., 2015)

Possibly related to the scarce use of biomass burning for domestic heating in the Mediterranean region compared to Central and Northern Europe, very few studies have been published describing BrC effects on intensive aerosol optical properties in the WMB. However, recent studies have estimated that biomass burning sources in the WMB may contribute more than expected to the measured ambient elemental carbon (EC) and organic carbon (OC) concentrations (Minguillón et al., 2011 and 2015; Reche et al., 2012; Mohr et al., 2012; Viana et al., 2013; Pandolfi et al., 2014b). In these studies the biomass burning source was characterized by means of techniques such as positive matrix factorization (PMF) on AMS (Aerosol Mass Spectrometer) or ACSM (Aerosol Chemical Speciation Monitor) data, filter-based analysis of 14C and/or specific chemical tracers such as levoglucosan or K+. Nevertheless, only few studies have used multi-wavelength Aethalometer data (Sandradewi et al., 2008b) in the WMB (Segura et al., 2014).

The main aim of this work is to provide a deep characterization of the intensive optical properties of atmospheric aerosols in the WMB under specific pollution episodes (SDE and BB). Thus, here we evaluate the feasibility of using the intensive aerosol optical properties for the near real-time detection of specific atmospheric events in the WMB. A sensitivity study aimed at calibrating the measured intensive aerosol optical properties is presented and discussed. We show that this calibration is needed to take into account the effects of local pollution on the intensive optical properties during SDE and BB

events. Moreover, we provide the range of variability of the calculated intensive optical properties as a function of the intensity of these events. This information is a valuable input for models studying the radiative effects of atmospheric aerosols in this very peculiar area. With this aim we used high-quality data collected at two stations located in the WMB: Montseny (MSY, regional background station; 720 m a.s.l.) and Montsec (MSA, remote station; 1500 m a.s.l.).

## 2. Methodology

### 2.1 Sampling sites

Results presented in this study were obtained from data collected at two in situ measurement stations located in the NE Iberian Peninsula (Fig. 1). The Montseny (MSY) site, integrated in the ACTRIS (Aerosol, Clouds and Trace gases Research InfraStructure) network, is a middle altitude emplacement (720 m a.s.l.) representative of the regional background in the WMB. The MSY measurement station is located in the Montseny Natural Park (41°19'N, 02°21'E), 40 Km to the NNE of the Barcelona urban area and 25 km from the Mediterranean coast, and is frequently affected by anthropogenic emissions (Pérez et al., 2008). The continental background site Montsec (MSA), integrated in the GAW (Global Atmosphere Watch) programme, is a remote high altitude emplacement (1570 m a.s.l.) situated in the southern side of the Pre-Pyreness at the Montsec d'Ares mountain (42°3'N, 0°44'E), 140 km to the NW of Barcelona and 140 Km to the WNW of Montseny (Ripoll et al., 2014). In situ optical aerosol properties measured at these two sites were performed following the standards required by GAW and ACTRIS networks.

Detailed information on these monitoring stations can be found for example in Pérez et al. (2008), Pey et al. (2009), Pandolfi et al. (2011), Cusack et al. (2012), and Minguillón et al. (2015) for MSY, and in Ripoll et al. (2014, and 2015b), Pandolfi et al. (2014a) for the MSA site.

### 2.2 Classification of atmospheric scenarios

The classification of atmospheric episodes affecting MSA and MSY sites on each day of the sampling period was performed following the procedure described by Ripoll et al. (2014) using BSC-DREAM8b (Basart et al., 2012), SKIRON (Nickovic et al., 2001) and HYSPLIT (Draxler and Rolph, 2015; Rolph 2015) models.

A detailed description of the main meteorological processes affecting the area under study can be found in Pérez et al., (2008); Pey et al., (2010); Pandolfi et al., (2014a); Ripoll et al., (2014). This study is focused in the atmospheric scenarios affecting significantly the concentrations of pollutants in the WMB: North African (NAF), summer regional (REG) and Atlantic advections (AA). SDE, driven by NAF air masses, are more frequent from March to October strongly contributing to increase $PM_{10}$. The summer REG scenarios favour the dispersion of the pollutants around the emission sources and the transport and accumulation of pollutants through the regional recirculation of air masses (Millán et al., 1997). Often REG occur after SDE causing important effects on air quality as shown later. Atlantic advections (AA) affect the WMB throughout the year but mainly in winter. Fresh and clean air masses from the Atlantic clear out the previously accumulated

stagnated air masses, leading to lower pollutant concentrations at regional scale. The seasonal distribution of the main atmospheric episodes throughout the year is very similar at MSY and MSA. However, during colder periods MSA high altitude station is frequently within the free troposphere conditions whereas MSY station is frequently affected by regional/local emission sources being often within the planetary boundary layer PBL (Pandolfi et al., 2014 a, b).

The African dust contribution to $PM_{10}$ (%dust) at MSY was calculated by the statistical methodology described in Escudero et al. (2007b) and Pey et al. (2013). This method is based on the application of 30 days moving 40th percentile to the daily $PM_{10}$ data series, after excluding those days impacted by African dust. For those days affected by African dust the percentile value is assumed to be the theoretical background concentration of PM if African dust did not occur. After that, the African dust daily contribution is obtained as the difference between the experimental $PM_{10}$ concentration and the calculated 40th percentile value.

## 2.3 Measurements and instrumentation

### 2.3.1 Aerosol absorption and Equivalent black carbon (BC) concentration measurements

Aerosol light absorption coefficient ($\sigma_{ap}$) at 637 nm (Müller et al., 2011a) was measured at 1 min resolution with a Multi Angle Absorption Photometer (MAAP, model 5012, Thermo). BC mass concentrations (Petzold et al., 2013) were calculated assuming a constant mass absorption cross section (MAC) of 6.6 $m^2$ $g^{-1}$ (Petzold and Schönlinner, 2004). The detection limit of the MAAP instrument is lower than 100 ng $m^{-3}$ over 2 min integration.

Aerosol light absorption coefficients ($\sigma_{ap}$) at seven different wavelengths (370, 470, 520, 590, 660, 880 and 950 nm) were obtained every 1 min at both stations by means of Aethalometer instruments (models AE-31 and AE-33). At MSA site the AE-33 (Drinovec et al., 2015) was equipped with a $PM_{2.5}$ cut-off inlet until March 2014 and with a $PM_{10}$ cut-off inlet afterwards. Absorption measurements at MSY station were carried out with a $PM_{10}$ cut-off inlet using an AE-31 Aethalometer model from June 2012 to June 2013, then replaced with an AE-33 model. Absorption measurements from the AE-31 were corrected for loading and scattering effects according to Weingartner et al. (2003). The site-specific AE-31 multiple scattering correction factor (C) at MSY was obtained by comparing with measurements from MAAP and it was estimated in around 3.6. Data was normalized to standard conditions (273K, 1013 hPa). Multi-wavelength aerosol absorption measurements used in this work cover a period of 2.5 year at MSY (June 2012-December 2014) and around 1 year at MSA (November 2013-December 2014).

### 2.3.2 Aerosol scattering measurements

Aerosols light scattering ($\sigma_{sp}$) and hemispheric backscattering ($\sigma_{bsp}$) coefficients were measured at each site every 5 min at three different wavelengths (450, 525 and 635 nm) with a LED-based integrating nephelometer (model Aurora 3000, ECOTECH Pty,Ltd, Knoxfield, Australia). Calibration of the nephelometer was performed three times per year by using $CO_2$ as span gas while zero adjusts were performed once per day by using internally filtered particle free air. A relative humidity (RH) threshold was set following the ACTRIS recommendations (RH<40%). Scattering measurements were corrected for

truncation due to non-ideal detection of scattered radiation following the procedure described in (Müller et al., 2011b). Multi-wavelength aerosol scattering measurements used in this work cover a period of 5 years at MSY (from January 2010 to December 2014) and 3.5 years at MSA (from July 2011 to December 2014).

#### 2.3.3 PM measurements

Real-time PM concentrations were continuously measured at 30 and 5 min resolution by optical particle counters (OPC) using GRIMM spectrometers (GRIMM 180 at MSY, and GRIMM 1107 and GRIMM 1129 at MSA). Concentrations were corrected by comparison with 24 h gravimetric mass measurements of PMx (Alastuey et al., 2011). For gravimetric measurements 24h PMx samples were collected every 4 days on 150 mm quartz micro-fiber filters (Pallflex QAT) with high-volume (Hi-Vol) samplers (DIGITEL DH80 and/or MCV CAV-A/MSb at 30 $m^3$ $h^{-1}$).

## 3. Calculation of the intensive aerosol optical properties

The extensive and intensive aerosol optical properties and the equations used to derive the intensive properties are reported in Table 1 and briefly commented below.

In order to study some of the aforementioned intensive optical properties over a wider spectral range, the 3λ scattering measurements from nephelometer were derived at the 7 Aethalometer wavelengths using the SAE calculated from 3λ measured scattering. Once scattering was obtained at the 7λ, we estimated SSA and SSAAE at these 7λ.

The extensive and intensive aerosol optical properties and the equations used to derive the intensive properties are reported in Table 1 and briefly commented below.

In order to study some of the aforementioned intensive optical properties over a wider spectral range, the 3λ scattering measurements from nephelometer were derived at the 7 Aethalometer wavelengths using the SAE calculated from 3λ measured scattering. Once scattering was obtained at the 7λ, we estimated SSA and SSAAE at these 7λ.

a) The SAE depends on the physical properties of aerosols and mainly on the size of the particles. Generally, SAE lower than 1 or higher than 2 indicate that the scattering is dominated by larger or finer particles, respectively (Seinfeld and Pandis, 1998; Schuster et al., 2006). In this study SAE was estimated from a linear fit of 3λ scattering measured in the 450-635 nm range.

b) The *g* parameter (Delene and Ogren, 2002; Andrews et al., 2006) is defined as the cosine-weighted average of the phase function which is the probability of radiation being scattered in a given direction. Values of *g* can range from -1 for 180° backwards scattering to +1 for complete forward scattering (0°). A value of 0.7 is commonly used in radiative transfer models (Ogren et al., 2006).

c) The AAE provides information about the chemical composition of atmospheric aerosols. BC absorbs radiation in the whole solar spectrum with the same efficiency, thus it is characterized by AAE values around 1 (Kirchstetter et al., 2004; Kim et al., 2012). Conversely, BrC and mineral dust show strong light absorption in the blue to ultraviolet

spectrum leading to AAE values up to 3 and 6.5 respectively (Kirchstetter, 2004; Chen and Bond, 2010; Kim et al., 2012; and Petzold et al., 2009). AAE was estimated from a linear fit of 7λ absorption measured in the 370-950 nm range.

d) The SSA parameter is defined as the ratio between the scattering and the extinction coefficients at a given wavelength and describes the relative importance of scattering and absorption on radiation. Thus the SSA parameter indicates the potential of aerosols for cooling or warming the atmosphere. A detailed description of SSA at both MSY and MSA was presented by Pandolfi et al., (2011) and (2014a), respectively. Nevertheless in this work the SSA is used with the main objective of calculating SSAAE.

e) The wavelength dependence of the SSA is known as the SSAAE and it is defined as SSAAE=(1-SSA)*(SAE-AAE) (Moosmüller and Chakrabarty, 2011). This parameter provides general information about the type of sampled aerosols integrating both physical and chemical properties, and it has been proposed as a good indicator for the presence of Saharan dust in the atmosphere (Collaud Coen et al., 2004). The Saharan dust outbreaks change the intensive optical properties of sampled aerosols causing a reduction of SAE and an increase of AAE, resulting in a negative SSAAE during these events. Therefore this parameter can be used to asses which type of aerosol is dominating the scattering and the absorption. For example Collaud Coen et al. (2004) reported measurements performed at the high altitude alpine station Jungfraujoch (Switzerland) and showed that the SSAAE was able to detect 100% of Saharan dust outbreaks compared with 80% and around 40% of events detected using SAE and AAE, respectively. Other works have used SSAAE to distinguish between the two important sources of UV absorbing aerosols, biomass burning and Saharan dust, as is detailed in (Russell et al., 2010). The SSAAE was estimated from a linear fit of 7λ-SSA calculated in the 370-950 nm range (Table 1).

## 4. The Aethalometer model

The Aethalometer (AE) model allows the detection of fossil fuel combustion (FF) and biomass burning (BB) contributions to the total BC concentrations taking advantage of the different spectral absorption efficiency of the main markers of these two sources: BC for FF combustion and BrC for BB (Sandradewi et al., 2008b). The AE model has also been applied for FF and BB source apportionment to total carbonaceous material ($CM_{total}$=OM+BC) and to organic matter (OM) (Favez et al., 2010). Light absorption measurements at 370-450 nm and 880-950 nm are used due to the fact that BC from FF combustion has a weak dependence on wavelength whereas BrC from BB shows enhanced absorption at shorter wavelengths. Here we applied the AE model to absorption measurements performed at 370 nm and 950 nm.

The AE model is usually applied selecting AAE values around 0.8-1.1 for BC from FF combustion ($AAE_{ff}$) and around 1.6-2.2 for BB ($AAE_{bb}$). It is known that the AE method may lead to high uncertainties in the estimation of biomass burning contribution due to the high variability of $AAE_{bb}$ depending on the wood burned combustion regime and on the internal mixing with non-absorbing materials (Lewis et al., 2008; Harrison et al., 2013). Thus, $AAE_{ff}$ and $AAE_{bb}$ are usually chosen

by comparing the AE model outputs with FF and BB contributions to BC and/or OM from other techniques such as chemical mass balance (CMB) model on off-line filter measurements, positive matrix factorization (PMF) model on AMS and/or ACSM data or $^{14}C$ technique (Favez et al., 2010; Herich et al., 2011; Crippa et al., 2013). Here we followed a similar procedure to calibrate the AE model: the optimal $AAE_{ff}$ and $AAE_{bb}$ were selected comparing results from the AE model with those obtained from PMF on simultaneous ACSM hourly data at MSY station for 1 year (Minguillón et al., 2015). Then, the optimal $AAE_{ff}$ and $AAE_{bb}$ for MSY were applied to MSA Aethalometer model.

In this work, $CM_{total}$ was calculated as the sum of BC concentration measured by MAAP (637 nm) and OM measured by ACSM. Following equations (1-3), $CM_{total}$ was expressed as the sum of carbonaceous material from FF combustion ($CM_{ff}$), carbonaceous material from BB emissions ($CM_{bb}$) and non-combustion organic aerosols (OA). At MSY station, OA may account for a large contribution mainly in summer and includes principally organic aerosols from biogenic origin as reported in Minguillón et al. (2011) and Pandolfi et al. (2014b). Thus, we included the constant $C_3$ in contrast to previous studies where it was negligible assuming a low contribution of OA sources. $CM_{ff}$ and $CM_{bb}$ were then expressed as the product of the constants ($C_1$ and $C_2$) multiplied by the aerosol absorption due to FF at 950 nm ($b_{abs,ff,950}$) and the aerosol absorption due to BB at 370 nm ($b_{abs,bb,370}$), respectively. The $b_{abs,ff,950}$ and $b_{abs,bb,370}$ were calculated for different values of $AAE_{ff}$ and $AAE_{bb}$ following the equations reported in Sandradewi et al. (2008b) and then used in eqs. 1-3 for OM source apportionment. Finally, the constants $C_1$, $C_2$ and $C_3$, which related the light absorption to the particulate mass, were calculated by multilinear regression (MLR) analysis.

$$CM_{total} = CM_{ff} + CM_{bb} + OA \quad (1)$$

$$OM + BC = C_1.b_{abs,ff,950} + C_2.b_{abs,bb,370} + C_3 \quad (2)$$

$$OM + BC = (OM_{ff}+BC_{ff})_{950} + (OM_{bb}+BC_{bb})_{370} + OA \quad (3)$$

Once $BC_{ff}$, $BC_{bb}$, $CM_{ff}$ and $CM_{bb}$ have been estimated, the contributions of FF and BB to OM ($OM_{ff}$ and $OM_{bb}$) can be calculated by subtracting BC to CM (Favez et al., 2010).

## 5. Results and discussion

### 5.1 General features

Mean, standard deviation, median, minimum, maximum, skewness and percentiles (5, 25, 50, 75, 95) of hourly extensive and intensive aerosol optical properties used in this work are reported in Tables S1a and S1b. Although the periods considered at the two stations were different, time coverage was sufficiently large to allow for a characterization of the mean aerosol optical properties at the two sites. Mean values of scattering, backscattering and $PM_{10}$ concentrations at both sites were consistent with previous studies performed at these stations (Pandolfi et al., 2011, 2014a; Ripoll et al., 2014, 2015b). Higher $\sigma_{sp}$ and $\sigma_{bsp}$ were on average measured at MSY consistent with higher $PM_{10}$ concentrations due to the larger impact of anthropogenic sources at this station. Consequently, larger absorption $\sigma_{ap}$ ($Mm^{-1}$) at 470 and 880 nm was also observed at MSY (7.66±6.5 and 3.51±2.99) compared to MSA (3.57±3.95 and 1.59±1.71).

Mean values of *g* (525 nm), SAE and AAE at MSY station were 0.59±0.06; 1.38±0.79 and 1.30±0.30, respectively. At MSA station mean values for these parameters were 0.57±0.14, 1.58±0.83 and 1.36±0.27. Mean SAE was higher at MSA station compared to MSY which could be explained by a dominance of smaller particles on average at MSA likely due to frequent position of the station within the free troposphere in winter. As already reported (Andrews et al., 2011; Berkowitz et al., 2011; Marcq et al., 2010; Pandolfi et al., 2014a), under low aerosol loadings at mountain top sites the aerosol mixture is preferentially composed of relatively smaller (and darker) particles. Despite the lesser variability of AAE with respect to other parameters, MSY site presented slightly lower values due to a major predominance of black carbon particles as a consequence of the proximity to Barcelona urban area. SSA
was slightly higher at MSA (0.85±0.08 and 0.82±0.3) compared to MSY (0.83±0.07 and 0.8±0.12) at 470 and 880 nm, respectively.

### 5.2 Detection of Saharan dust outbreaks using aerosol intensive optical properties

As already observed, SDE can be detected using optical properties measurements taking advantage of the changes that mineral dust causes in the spectral dependence of aerosol scattering and absorption (Collaud Coen et al., 2004). In fact SDE scenarios are characterized by a decrease of SAE, as a consequence of the predominance of coarse particles, and an increase of AAE due to the enhanced absorption in the UV spectrum by mineral dust. Therefore, the scatter plot between AAE and SAE (called Ångström matrix) is useful to detect periods dominated by SDE (Russell et al., 2010). Here we calibrate, based on the available tools, the Ångström matrices for MSY and MSA in order to use them for SDE detection.

The Ångström matrix for MSY and MSA (Fig. 2b, e) showed dominance of coarse material (high % of $PM_{1-10}$ in $PM_{10}$) related to low values of SAE (roughly lower than 1) and larger values of AAE (approximately higher than 1.3) during SDE. In order to demonstrate that these SAE and AAE limits were mainly related with the presence of mineral dust from Africa in the area under study, the Ångström matrices were also weighted by the occurrence of the three main atmospheric scenarios affecting MSY and MSA stations: SDE, REG and AA (Figs. 2a, d). As shown in Figs. 2a, d the region of the Ångström matrices representing SDE well fits with the SAE and AAE limits reported above (Figs. 2b, e).

The feasibility of detecting Saharan dust outbreaks by means of the hourly Ångström matrices is further confirmed in Fig. S1, where the Ångström matrix for MSY station was weighted by the %dust (daily base) for those days affected by SDE. The quantification of African mineral dust contribution to $PM_{10}$ (%dust) at MSY was calculated by the statistical methodology described in Escudero et al. (2007b) and Pey et al. (2013a) (detailed in sub section 2.2). Despite the scarce availability of simultaneous daily data points of SAE, AAE and %dust for the period under study, the Ångström matrix showed lower SAE and increasing AAE with increasing intensity of SDE (%dust), in agreement with the Ångström matrix reported in Fig. 2b. However, Fig. S1 clearly shows that there are conditions when the AAE-SAE pair does not unequivocally detect the Saharan dust outbreaks, being SAE higher than 1.0-1.5 and AAE lower than 1.2-1.3. These points are characterized by relatively low (<40% approximately) dust contribution to $PM_{10}$ representing not very intense SDE. Thus, this region of the Ångström matrix identified an aerosol mixture between mineral dust and anthropogenic pollutants of

mainly local origin. Then, we can conclude that: a) some points during REG episodes (yellow dots in Figs. 1a, d) were characterized by SAE and AAE values similar to those observed during SDE indicating presence of mineral dust in the atmosphere, and that b) for some SDE, the corresponding AAE-SAE pairs do not unequivocally confirm the presence of mineral dust (anthropogenic emissions and mineral dust mixing).

The blue spot area displayed in the Ångström matrix for MSA station (Fig. 2e) showed AAE-SAE pairs characterized by low contribution of $PM_{1-10}$ to $PM_{10}$~%1-10, which are mainly represented by AA scenarios. These AA scenarios, some of them related with free troposphere conditions in MSA during winter, lead to a cleaner environment free of pollutants characterized by finer and relatively darker particles in the Ångström matrix. Conversely, a predominance of REG scenarios is seen at MSY (yellow dots in Fig. 1a), related to larger contribution of $PM_{1-10}$ to $PM_{10}$ (40-80%) (Fig. 2b). REG episodes, mainly related to pollution scenarios, are characterized by local (affecting lower altitude regions driven by the breeze patterns) to regional (reaching higher altitude locations driven by larger circulations and upslope winds) atmospheric circulations transporting fine particles from the urbanized/industrialized coastline. Mean SAE ranged between 1.5-3 and 1.3-2.3 at MSY and MSA stations, respectively, during REG scenarios, whereas main AAE values ranged between 1-1.7 at both stations. Recently, Mallet et al. (2013) reported column-integrated AAE (440-870 nm) values across the Mediterranean using Level 2 AERONET data varying from around 1.3 in urban areas to more than 2 at Mediterranean dusty sites.

In order to study how SDE affect the asymmetry parameter in the area under study, Figs. 2c, f show a modified Ångström matrix where the *g* parameter was investigated instead of SAE at both stations. This parameter can also be used to estimate the size of aerosols according to the difference in the scattering direction presented by small and larger particles, since larger particles present higher forward than backward scattering. During SDE, *g* was similar at both stations, approximately ranging between 0.55-0.75 at MSA and between 0.5-0.7 at MSY. These results are in agreement with those g values reported by Ogren et al. (2006) for other in situ measurements. Therefore, given that SAE parameter presents larger variability than *g* in relation to changes in $\%PM_{1-10}$, we conclude that SAE is a better proxy for estimating aerosol size. Despite this, providing experimental variability ranges for *g* is important given that the asymmetry parameter is commonly used in radiative transfer models (Ogren et al., 2006).

As already mentioned, the SSAAE has been identified as a good indicator for Saharan dust outbreaks at mountain top sites being negative during these types of events (Collaud Coen et al., 2004). The SSAAE is a useful parameter, which can be used together with the Ångström matrix in order to characterize mineral dust at different emplacements with the aim to identify SDE in real-time. Similarly to what already observed for the Ångström matrices, our results showed that the feasibility of detecting SDE by means of SSAAE depended on both the location and altitude of the measurement station, which determines the aerosol background concentration, and the intensity of the SDE.

Figure 3a showed a relationship between SSAAE and %dust at MSY for those days affected by SDE. At MSA, where %dust was not calculated due to limitations of the methodology, SSSAE did correlate with percentage of coarse particles in $PM_{10}$ (Fig. 3b). SSAAE became negative for most of the SDE identified at MSA accounting for 85% detection of these events. However SSAAE showed more frequently positive values near to zero at MSY, detecting 50% of SDE due to a larger

exposure to anthropogenic emissions. The SSAAE became negative when the relative contribution of Saharan dust to $PM_{10}$ (%dust) at MSY was higher than approximately 60%, keeping positive values at lower %dust in $PM_{10}$ despite the presence of mineral dust.

Figure 3c shows an example of the daily variation of SSAAE, SAE and AAE at MSY during a SDE. Low values of SAE (<1) and higher values of AAE (>1.5) lead to negative SSAAE during the night, indicating presence of mineral dust. Conversely, during the day, anthropogenic fine pollutants transported from nearby polluted areas hindered the optical effect of mineral dust during non-intense SDE (54% of dust in $PM_{10}$). Consequently, despite the impact of mineral dust, the SSAAE turned into positive values. SAE reached values around 2 indicating dominance of fine particles and, correspondingly, the AAE lowered to around 1.2 indicative that these fine particles were mainly of anthropogenic origin. Thus, the proximity to anthropogenic sources under specific atmospheric conditions (i.e. strong breeze and low SDE intensity) can prevent both the Ångström matrix and the SSAAE parameter from detecting SDE.

A different scenario is shown in Fig. 3d, where two Saharan dust outbreaks were detected and highlighted by the yellow rectangles. The SSAAE was negative during the two outbreaks keeping negative values between the two events despite the influence of Atlantic air masses during the days 23th and 24th October 2013. Interestingly, the SSAAE reached the lowest negative values during the subsequent days after the SDE, until precipitation scavenged pollutants from the atmosphere (highlighted by the blue rectangle). Thus, the local and regional recirculation of air masses under the REG episode, often lasting for a few days, recirculated an aerosol mixture dominated by coarse Saharan particles in the atmosphere at a level able to cause the SSAAE be negative even in absence of African air mass advection (Fig. S2). The evidence that mineral dust can recirculate under dry conditions in summer for a few days after the SDE is of high relevance for air quality. Thus, near-real-time aerosol optical parameters such as SSAAE are very useful to detect mineral dust in the atmosphere even after the end of the event.

## 5.3 Detection of biomass burning using aerosol optical properties

### 5.3.1 Calculation of the constants from the Aethalometer model

In order to test the stability of the AE model for our emplacement (MSY), $C_1$, $C_2$ and $C_3$ were calculated varying (Table 2): a) $AAE_{bb}$ between 1.8 and 2.2 (for a fixed $AAE_{ff}$=1), and b) $AAE_{ff}$ between 0.9 and 1.1 (for a fixed $AAE_{bb}$=2). In the first case (a) $C_1$ showed a very low variability keeping values around 1.05±0.01 g $m^{-2}$ ,whereas $C_2$ showed a higher variability ranging between 0.28 g $m^{-2}$ ($AAE_{bb}$=1.8) to 0.24 g $m^{-2}$ ($AAE_{bb}$=2.2). In our work $C_3$, which represents the contribution from non-combustion OM, was estimated in around 0.31±0.04 µg $m^{-3}$. These results were consistent with previous studies dealing with AE source apportionment to OM and reporting less variability for $C_1$ compared to $C_2$ (Sandradewi et al., 2008b; Favez et al., 2010). In another study (Herich et al., 2011) the AE model was not applied to OM mainly due to the high variability (i.e. model instability) observed for $C_1$ from different model outputs. In the second case (b), $C_1$ changed only little (less than 10%) ranging between 1.01 g $m^{-2}$ ($AAE_{ff}$=0.9) to 1.09 g $m^{-2}$ ($AAE_{ff}$=1.1) for a fixed $AAE_{bb}$ of 2. As reported bellow, $AAE_{bb}$ for our environment was set to 2 by comparison with ancillary experimental measurements, whereas $AAE_{ff}$ was set to 1 as in

previous studies, given the lower sensitivity of the AE model to $AAE_{ff}$ compared to $AAE_{bb}$. It is important to consider that the values of $C_1$ (~1.05 g $m^{-2}$) and $C_2$ (~0.26 g $m^{-2}$) calculated for our emplacement were different from those reported in previous studies for different environments. In their works, Favez et al. (2010; Grenoble) and Sandradewi et al. (2008b; Roveredo, Switzerland) set $C_1$ to a fixed value of 0.26 g $m^{-2}$, being this parameter less variable, and $C_2$ was estimated around 0.7-0.8 g $m^{-2}$. Differences between the constants were due to the larger use of biofuel for domestic heating in these later locations, leading to higher contribution of BB to BC compared to FF combustion sources (and probably less effect of FF sources). Contrary to our emplacement where results indicated (as shown later) higher contribution from FF sources compared to BB for both BC and OM.

Given the large differences our constants $C_1$ and $C_2$ showed compared to previous studies for different environments, we applied here a similar procedure as described in Herich et al. (2011). Thus, we simulated $CM_{total}$ using $C_1$ and $C_2$ from Sandradewi et al. (2008b) and Favez et al. (2010), and $b_{abs,ff,(\lambda 1)}$ and $b_{abs,bb,(\lambda 2)}$ as derived from our measurements. As expected the results showed very low correlation between calculated and measured CM ($R^2$=0.009; slope=0.65) compared to $R^2$=1 and slope=1 using our calculated constants $C_1$, $C_2$ and $C_3$. Therefore we conclude that calculation of the specific constants of the model for the area under study is required in order to successfully perform the Aethalometer model.

Moreover, we calculated $C_1$, $C_2$ and $C_3$ for two more different cases: (a) including only the winter season in order to account for a larger contribution of BB emissions and to reduce the influence of non-combustion OM and SOA formation which maximize in summer at MSY station (Minguillón et al., 2011), and (b) excluding SDE from the database which could overlap with BrC being both, BB and mineral dust, important absorbers in the UV. The differences for $C_1$, $C_2$ and $C_3$ calculated between these two cases and the whole period (June 2012–July 2013, Table 2) in case (a) were lower than 10%, 20% and 15%, respectively. These differences were around 3%, 6% and 34%, respectively, for the case (b). Given that the AE model outputs have been estimated having errors as high as 50% (Favez et al., 2010) and given that we are continuously measuring absorption with the AE instrument at MSY and MSA without ACSM data, the model was calibrated using 1 year data set in order to apply the AE model at any other period without ancillary measurements.

#### 5.3.2 Validation of the Aethalometer model with simultaneous experimental data

Very few studies have been published comparing outputs from the AE model with the source apportionment of the ACSM measurements (Favez et al., 2010). Biomass burning organic aerosol (BBOA) and hydrocarbon-like organic aerosol (HOA) from ACSM data refer to primary organic aerosols (POA) whereas $OM_{bb}$ and $OM_{ff}$ from AE model include SOA formed from these primary sources. SOA formation from biomass burning emissions can be up to 25% of the BBOA emitted, as shown by Cubison et al. (2011) using Aerosol Mass Spectrometer (HR-ToF-AMS) data. Moreover, based on the results from the DAURE campaign carried out in March 2009 at MSY station, the organic carbon (OC) originated from fossil sources is only 15% primary at MSY (Minguillón et al., 2011), which corresponds to 10% if OM is considered instead of OC. Thus, we assume that primary BBOA and HOA represent approximately 75% and 10% of the $OM_{bb}$ and $OM_{ff}$, respectively, at MSY station.

Relationships between BBOA and $OM_{bb}$ concentration for different $AAE_{bb}$ values (Table 3) showed good agreement ($R^2$~0.43) with slopes ranging between 1.1 and 2.2 depending on the $AAE_{bb}$ used. The relationship between $OM_{ff}$ and HOA showed less variable slope (F) (around 4) but more variable $R^2$ between 0.43 and 0.63. Choosing $AAE_{bb}$=2 and $AAE_{ff}$=1 we obtained: a) an $OM_{bb}$/BBOA ratio of around 1.27 ($R^2$=0.43) in agreement with 25% of SOA formation from primary biomass burning emissions estimated by Cubison et al. (2011); and b) an $OM_{ff}$/HOA ratio of 4.4 ($R^2$=0.6) which is consistent with 90% portion of SOA found at MSY in previous studies (Minguillón et al., 2011). The correlations were only moderate mainly due to the variable SOA formation, which is partially driven by the environmental conditions, as opposed to the primary OA emissions. Moreover, it should be noted that the slopes and $R^2$ in Table 3 were obtained using hourly averages. Scatterplots by bins (Fig 4) showed that the relationships had slopes in agreement with those reported in Table 3 but much higher $R^2$ (0.97).

The relationship between $OM_{bb}$ and BBOA calculated only for the winter period using hourly data showed $R^2$=0.4 and F=0.96. The slope was close to the unity due to the lower SOA formation in winter, consequence of less photochemistry activity, and the prevalence of primary emissions. Experimental measurements of Nitrogen dioxide ($NO_2$), which is mainly related to fossil fuel emissions, agrees well ($R^2$=0.64) with $BC_{ff}$ obtained from $AAE_{bb}$=2 and $AAE_{ff}$=1 for the winter period at MSY (Fig. 5).

Besides uncertainties in determining FF and BB contributions from the Aethalometer model, results from sensitivity test analysis showed good agreement with experimental measurements and good stability of the model. We have shown that the constants $C_1$, $C_2$ and $C_3$ depend on the relative contributions of FF and BB, thus these constants are site-dependent and should be calculated for each measurement emplacement. Moreover, a calibration of the model is necessary to determine the most suitable $AAE_{ff}$ and $AAE_{bb}$ pair for a reliable estimation of fossil fuel and biomass burning contributions. Interestingly $AAE_{bb}$ and $AAE_{ff}$ chosen in this work were the same as in other studies, suggesting a stable value of AAE=2 for characterizing BB emissions within the model. Our results showed that the higher $AAE_{bb}$ the lower the estimated $BC_{bb}$ contribution, which ranged between 35-45% depending on the $AAE_{bb}$ used (1.8-2.2).

#### 5.3.3 Seasonal and daily variation of fossil fuel and biomass burning contribution to BC and OM at Montseny and Montsec stations

Seasonal and daily AAE and relative contributions of BB and FF to BC (at both MSY and MSA) and to OM (at MSY only) from the Aethalometer model are shown in Fig. 6. Both environments are characterized by similar average PM chemical composition (Ripoll et al., 2015b), thus probably leading to similar mean values of AAE at MSY (1.30±0.30) and MSA (1.36±0.26) (Figs. 6a, e). Thus, $AAE_{ff}$ and $AAE_{bb}$ determined for MSY were used also for MSA.

MSY showed slightly lower AAE as a consequence of higher exposure to FF emissions sources compared to MSA. AAE at MSA and MSY showed larger values on average in winter suggesting a higher contribution of BrC. AAE monthly averages reached around 1.5 at both sites. Despite the fact that the lowest BC and OM concentrations were observed in winter the AAE showed the highest values indicating larger contribution of BB sources at both stations. It is interesting to note that on

average AAE was higher at MSY than at MSA during winter months (December-January) suggesting higher relative BB contribution at MSY compared to MSA in winter (Fig. 6e and S3a). This was likely due to the fact that MSA station is often above the polluted PBL in winter whereas MSY, located at lower altitude, is usually within the PBL and frequently affected by local pollutants accumulated under winter anticyclonic conditions (Pandolfi et al., 2014b; Ripoll et al., 2015b). Low values of AAE during the day and higher at night at both sites resulted mainly from the development of sea and mountain breezes, favouring the transport of anthropogenic pollutants from the urbanized/industrialized coastline and valleys to inland areas and leading to an increase of AAE during the warmest hours of the day (Fig. 6a).

The measured BC was well reproduced by the sum of $BC_{ff}$ and $BC_{bb}$ contributions from the AE model showing slightly overestimation, by 11% and 15% at MSY (Fig. 6c, g) and MSA (Fig. 6d, h), respectively, on annual average. However, measured OM is underestimated by the sum of $OM_{ff}$ and $OM_{bb}$ at MSY, due to the large contribution of carbonaceous material from non-combustion sources ($C_3$) during the warmer months (27%) (Fig. 6f). This difference was mainly driven by biogenic sources which are expected to have important contribution in our measurement emplacement, particularly in summer due to the SOA formation. Then $C_3$ time variation was well reproduced by the model showing larger contribution during the summer period. Nevertheless, based on the available previous studies performed at MSY (Minguillón et al., 2011 and 2015; Pandolfi et al., 2014b), $C_3$ contribution might be slightly underestimated due to possible apportionment within $OM_{ff}$ and/or $OM_{bb}$. It should also be note that some SOA UV absorbing compounds originated from anthropogenic sources, such as nitroaromatic compounds which are the major contributors to the light absorption of the toluene SOA (Laskin et al., 2015), may be partially apportioned within $OM_{bb}$, possibly resulting in an overestimation of this later.

Interestingly, a relationship was observed between AAE and the relative contribution of $BC_{bb}$ to BC concentrations at MSY and MSA (Fig. 7). AAE increased up to 1.5 when $\%BC_{bb}$ was higher than around 50% of the total measured BC. The intercept of the linear fit was 1.01 and 1.15 at MSY and MSA, respectively, pointing to BC from FF sources as main absorber in absence of biomass burning events. Therefore, we can clearly appreciate the effect of BrC from biomass burning on AAE even if the mean $BC_{bb}$ contributions (0.13 µg $m^{-3}$ and 0.06 µg $m^{-3}$) at MSY and MSA, respectively, to the total BC were quite low (36% and 40%). Mean $OM_{bb}$ concentration at MSY was 0.9 µg $m^{-3}$, accounting for a 30% contribution to total OM.

The prominent increase of FF contribution at MSY and MSA in summer, when both stations are within the PBL and dominated by similar atmospheric circulations, is in agreement with lower AAE values. Stronger summer recirculation processes which are strengthened by sea and mountain breezes favour the transport of pollutants toward regional areas inland. Daily variation of both BC and OM is mainly driven by FF combustion from Barcelona anthropogenic sources. The daily cycle is more pronounced at MSY as a consequence of the proximity to Barcelona Metropolitan Area and the lower altitude compared to MSA. Despite OM is mainly driven by biogenic sources during the summer period at MSY, significant FF contribution is registered during the warmest hours of the day (Fig. S3b). However BB sources time variation, from both BC and OM, are leaded by local atmospheric processes as domestic heating turning into a dominant source during the colder months at both stations. Thus, during winter, $BC_{bb}$ and $BC_{ff}$ showed almost the same contribution reaching the maximum

values in the afternoon (Fig. S3c). Conversely $OM_{ff}$ daily cycle is decoupled from $OM_{bb}$, showing this later larger concentrations during the night given that it is mainly leaded by BB emissions from domestic heating emitted during the colder hours, and also possibly as a result of SOA formation after the OM was emitted (Fig. S3b). Note that during the night $OM_{bb}$ concentration does not present large variations, possibly because it remains as a residual layer above the thermal inversion.

FF contribution to OM and BC was found to be significant at MSY, according to the large values obtained for $C_1$ constant in the Aethalometer model. In order to compare the results with different source apportionment methods, the fossil fuel and non-fossil fuel contribution to EC ($EC_{ff}$, $EC_{non_ff}$) and OC ($OC_{ff}$, $OC_{non_ff}$) reported by Minguillón et al. (2011) by means of the $^{14}C$ technique at MSY for the periods February-March and July 2009 were taken as a reference. Given that AE measurements were not available at MSY during those periods, we averaged available contributions from the Aethalometer model for the same time-of-the-year periods during 2012, 2013 and 2014 for BC and during 2012 for OM. Despite the lack of overlapping in the dataset, results for BC contributions from both techniques ($^{14}C$ and AE model) showed good agreement. $BC_{ff}$ contributions calculated by the AE model in winter and summer were 53% and 73% respectively, whereas $EC_{ff}$ contributions derived from $^{14}C$ measurements accounted for 66% and 79%. However larger discrepancies were found for FF and BB contributions to OM. Results from the $^{14}C$ technique identified a FF contribution to OC of 31% and 25% for winter and summer, respectively, whereas the AE model resulted in a $OM_{ff}$ contribution of 39% and 58%, respectively. We also saw a $OM_{bb}$ contribution around twice more than OC non-fossil fuel. The apparently overestimation of $OM_{bb}$ and $OM_{ff}$, particularly in summer, compared to the available results from $^{14}C$ might be possibly leaded by the partially apportionment of non-combustion carbonaceous material and SOA anthropogenic within $OM_{bb}$ and/or $OM_{ff}$, as we commented above.

A second assessment of the AE model results was carried out by comparison with OA source apportionment results reported by Minguillón et al. (2015) for winter (28 October-7 April 2013) and summer (14 Juny-9 October 2012) at MSY based on ACSM measurements. The agreement needs to be evaluated considering the different outputs from each method; thus whereas the ACSM OA source apportionment identifies the contribution of primary fossil fuel (HOA) and biomass burning (BBOA) contributions, the AE model calculates the total (including the SOA) fossil fuel ($OM_{ff}$) and biomass burning ($OM_{bb}$) contributions. HOA contribution was 12% and 13% for winter and summer, whereas $OM_{ff}$ accounted for 47% and 59%. BBOA was identified only in winter with a contribution of 28%, and $OM_{bb}$ contribution was 37% for the same period. These results are in agreement assuming the ratios $OM_{ff}$-to-HOA and $OM_{bb}$-to-BBOA based on SOA-to-POA proportion, used in the previous section 5.4.3 in order to calibrate the Aethalometer model and fit the most suitable $AAE_{ff}$ and $AAE_{bb}$ representative of our environment.

An interesting wildfire episode detected at MSY took place the 23th of July 2012 with AAE increasing significantly up to 2 and the lowest value at 1.3 (Fig. 8). BB sources dominated BC and OM contributions accounting for 73% and 78% respectively, until the breezes were developed and transported pollutants from urban areas toward the station during the warmest hours of the day, resulting in a decrease of the AAE. As we shown previously for the whole dataset, good

agreement was found between measured and simulated BC. Conversely OM was slightly underestimated during the sunlight hours likely due to biogenic emissions and SOA formation by photochemical reactions.

## 6. Conclusions

The present work shows the variations of the intensive aerosol optical properties measured at regional (Montseny) and continental (Montsec) background stations in the WMB. We have studied the feasibility of using the near real-time optical measurements performed at these stations for the detection of specific atmospheric pollution episodes affecting the WMB: Saharan dust and biomass burning.

The Ångström matrix revealed that Saharan dust events (SDE) in the WMB were characterized by SAE on average lower than 1 due to the larger size of mineral dust particles and AAE values higher than 1.3 (up to 2.5 depending on the intensity of SDE) indicating absorption in the UV by iron oxide contained within the mineral dust. Linear relationships were found between AAE and increasing %dust at MSY (0.7) and $\%PM_{1\text{-}10}$ at MSA (0.4) confirming the enhanced absorption in the UV due to mineral dust from SDE. Interestingly, SAE showed higher sensitivity than *g* to characterize the size of aerosols, ranging this latter between 0.55-0.75 and 0.50-0.70 at MSY and MSA respectively during SDE.

Feasibility of detecting SDE by means of SSAAE depended on both the location and altitude of the measurement station, which determines the aerosol background concentration, and the intensity of the SDE. Better results were shown at higher altitude locations, at MSA were detected most of the SDE (85%), whereas at MSY, with a larger exposure to anthropogenic pollutants, the detection of SDE depended mainly on the intensity of the Saharan dust outbreak. At MSY site 50% of SDE were detected, which were unequivocally identified when the relative contribution of mineral dust to $PM_{10}$ was higher than 60%.

The proximity to anthropogenic sources of mainly fine particles can prevent both the Ångström matrix and the SSAAE parameter from detecting SDE. We have shown that transport of anthropogenic pollutants (mainly finer particles and precursors) from the urbanized/industrialized coastline towards regional areas inland can hinder the effect of mineral dust on the intensive aerosol optical properties during less intense SDE. We have also shown that regional atmospheric circulations occurring after SDE may favour the resuspension of mineral dust at regional level in the WMB. Thus mineral dust can remain in the atmosphere for a few days after the SDE. This fact is highly relevant for air quality since SDE frequently promote exceedances in the $PM_{10}$ daily limit value.

A sensitivity test performed to the Aethalometer model at MSY showed that the model constants, which are representative of the main emission sources, are actually site-dependent and should be calculated for the area under study. FF sources showed larger contribution than BB at MSY, leading to $C_1$=1.05 and $C_2$=0.26 (g $m^{-2}$) for $AAE_{ff}$=1 and $AAE_{bb}$=2. Moreover $C_3$ was found to be significant mainly due to the large contribution of biogenic sources at MSY, showing values around 0.31 (µg $m^{-3}$). Linear relations were found for comparisons between $OM_{bb}$ vs. BBOA ($R^2$=0.43) and $OM_{ff}$ vs. HOA ($R^2$=0.6) showing fitting slopes of 1.27 and 4.4 respectively, which are consistent with SOA formation from BB and FF (25% and 90%)

emissions. Results from these comparisons were used in order to calibrate the Aethalometer model, pointing to $AAE_{bb}$=2 and $AAE_{ff}$=1 as the most suitable values for our emplacement.

Annual averages of $BC_{bb}$ contributions at MSY (36%) and MSA (40%) were significantly lower compared to other studies in northern Europe, due to a lesser use of biomass burning as heating system. $OM_{bb}$ contributions accounted for 30%. BB source contribution to both BC and OM were predominant during winter, with increasing AAE up to 1.5 when $\%BC_{bb}$ was higher than 50%. Nevertheless, BC and OM were leaded by FF emissions sources during the summer period, due to stronger summer recirculation processes which are strengthened by sea and mountain breezes favouring the transport of pollutants toward regional areas inland. An interesting wildfire episode showed AAE values up to 2, accounting for BB contributions to BC and OM of 73% and 78% respectively.

The Aethalometer model is a powerful tool to reproduce long periods of real-time FF and BB contribution to BC, even in those areas where there is a predominance of carbonaceous material from non-combustion sources and BB emissions does not present very large contributions. BC, as the sum of $BC_{ff}$ and $BC_{bb}$, was well reproduced showing a slightly overestimation of 11% and 13% at MSY and MSA. Results for $BC_{ff}$ and $BC_{bb}$ in winter and summer were in agreement with previous studies at MSY deployed by $^{14}C$ analysis. Furthermore $BC_{ff}$ and $NO_2$, both representative of traffic sources, showed good correlation for the winter period ($R^2$=0.64).

However, the model presents larger uncertainty concerning OM apportionment as reported in other studies (Favez et al., 2010; Herich et al., 2011). Biogenic sources, which present important contribution in our emplacement, are probably slightly underestimated by the model due to the partially apportionment of $C_3$ constant within $OM_{ff}$ and $OM_{bb}$. Furthermore $OM_{bb}$ might be slightly overestimated due to the account of anthropogenic SOA within it, which can overlap the absorption in the UV range. Despite the uncertainties associated to the source apportionment technique, OM time variation appears to be well reproduced. Nevertheless, it should be taken into account OM formation and transformation processes occurring in the NWM at the time of performing the AE model results, where important photochemical reactions take place leaded by large anthropogenic emissions and high insolation (mainly in summer). Then, further research is needed in the identification of BrC emissions sources and their effects on optical properties, in particular for SOA formation and transformation processes.

We have demonstrated the potential of in situ aerosol optical measurements, from both Nephelometer and Aethalometer instruments, for detecting specific air pollution scenarios in near real-time. This is possible given the high sensitivity of particular intensive aerosol optical parameters to characterize different types of atmospheric aerosols. However, it is necessary to perform a previous sensitivity test in order to evaluate and calibrate the intensive optical properties for detecting specific pollution episodes at different emplacements.

## Acknowledgements

This work was supported by the MINECO (Spanish Ministry of Economy and Competitiveness), the MAGRAMA (Spanish Ministry of Agriculture, Food and Environment), the Generalitat de Catalunya (AGAUR 2014 SGR33 and the DGQA) and

FEDER funds under the PRISMA project (CGL2012-39623- C02/00). This work has received funding from the European Union's Horizon 2020 research and innovation programme under grant agreement No 654109. The authors would like to express their gratitude to D. C. Carslaw and K. Ropkins for providing the Openair software used in this paper (Carslaw and Ropkins, 2012; Carslaw, 2012).

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

**Table 1.** Extensive and intensive aerosol optical properties measured and derived, respectively, in this work.

| Extensive optical properties | | | | |
|---|---|---|---|---|
| Optical properties of $PM_{10}$ particles | symbol | λ [nm] | method | notes |
| Scattering | $\sigma^{\lambda}_{spPM_{10}}$ | 450, 525, 635 | Nephelometer[AURORA 3000 ECOTECH Pty,Ltd, Knoxfield, Australia] | Measurements corrected for truncation and non-Lambertian illumination function of the light source as in Müller et al. (2011b) |
| Backscattering | $\sigma^{\lambda}_{bspPM_{10}}$ | | | |
| Absorption | $\sigma^{\lambda}_{apPM_{10}}$ | 370, 470, 520, 590, 660, 880, 950 | Aethalometers model AE-31 and AE-33[MAGEE Scientific] | AE-31 measurements corrected for filter loading as in Weingartner et al. (2003) and Collaud Coen et al. (2010) |
| **Intensive optical properties** | | | | |
| Scattering Ångström exponent | *SAE* | 450 to 635 | $SAE = -Linear\ estimation\left(\frac{Ln(\sigma^{\lambda_1}_{spPM_{10}}\ to\ \sigma^{\lambda_3}_{spPM_{10}})}{Ln(\lambda_1\ to\ \lambda_3)}\right)$ | |
| Absorption Ångström exponent | *AAE* | 370 to 950 | $AAE = -Linear\ estimation\left(\frac{Ln(\sigma^{\lambda_1}_{apPM_{10}}\ to\ \sigma^{\lambda_7}_{apPM_{10}})}{Ln(\lambda_1\ to\ \lambda_7)}\right)$ | |
| Asymmetry parameter | *g* | 450, 525, 635 | $g(\lambda) = -7.14\left(\sigma^{\lambda}_{bsp}/\sigma^{\lambda}_{sp}\right)^3 + 7.46\left(\sigma^{\lambda}_{bsp}/\sigma^{\lambda}_{sp}\right)^2 - 3.96\left(\sigma^{\lambda}_{bsp}/\sigma^{\lambda}_{sp}\right) + 0.9893$ | The nephelometer measures hemispheric backscattering [-90º - + 90º] |
| Single scattering albedo | *SSA* | 370, 470, 520, 590, 660, 880, 950 | $SSA(\lambda) = \frac{\sigma^{\lambda}_{sp}}{\sigma^{\lambda}_{sp} + \sigma^{\lambda}_{ap}}$ | In order to estimate SSA at 7 λ, the scattering was calculated at 7 λ using the measured SAE. |
| Single scattering albedo Ångström exponent | *SSAAE* | 370 to 950 | $SSAAE = -Linear\ estimation\left(\frac{Ln(SSA^{\lambda_1}_{PM_{10}}\ to\ SSA^{\lambda_7}_{PM_{10}})}{Ln(\lambda_1\ to\ \lambda_7)}\right)$ | |

**Table 2.** $C_1$, $C_2$ and $C_3$ obtained by MLR on the Aethalometer model for different $AAE_{bb}$ (1.8, 2, 2.2) keeping $AAE_{ff}$=1, and varying $AAE_{ff}$ (0.9, 1.1) keeping $AAE_{bb}$=2 on hourly base at MSY.

| Hourly data (5456) | $AAE_{ff}$=1 | | |
|---|---|---|---|
| | $AAE_{bb}$=1.8 | $AAE_{bb}$=2 | $AAE_{bb}$=2.2 |
| $C_1$ (g $m^{-2}$) | 1.05153 ± 0.01004 | | |
| $C_2$ (g $m^{-2}$) | 0.27998 ± 0.00314 | 0.26021 ± 0.00354 | 0.24384 ± 0.00393 |
| $C_3$ (µg $m^{-3}$) | 0.31433 ± 0.04051 | | |
| | $AAE_{bb}$=2 | | |
| | $AAE_{ff}$=0.9 | $AAE_{ff}$=1.1 | |
| $C_1$ (g $m^{-2}$) | 1.01342 ± 0.01063 | 1.09341 ± 0.00959 | |
| $C_2$ (g $m^{-2}$) | 0.26021 ± 0.00354 | | |
| $C_3$ (µg $m^{-3}$) | 0.31433 ± 0.04051 | | |

**Table 3.** Squared Pearson ($R^2$) and slope (F) of the scatterplot between $OM_{bb}$ and BBOA and between $OM_{ff}$ and HOA, for different values of $AAE_{bb}$ (1.6, 1.8, 2, 2.2) keeping $AAE_{ff}$=1 at MSY (hourly base).

| | | $AAE_{ff}$=1 | | | |
|---|---|---|---|---|---|
| | $AAE_{bb}$ | 1.6 | 1.8 | 2 | 2.2 |
| $OM_{bb}$ vs. BBOA | $R^2$ | 0.436 | 0.423 | 0.429 | 0.426 |
| | F | 2.160 | 1.440 | 1.274 | 1.064 |
| $OM_{ff}$ vs. HOA | $R^2$ | 0.426 | 0.525 | 0.600 | 0.631 |
| | F | 4.002 | 4.240 | 4.377 | 4.467 |

## Figure captions

**Figure 1. (a)** Location of Montsec (MSA; remote-mountaintop site) and Montseny (MSY; regional background) measurement sites. **(b)** Topographic profile of MSA and MSY area.

**Figure 2.** Ångström matrix (scatterplot of AAE vs. SAE weighted by air mass origin) at **(a)** MSY and **(d)** MSA. Ångström matrix (scatterplot of AAE vs. SAE weighted by levels of $\%PM_{1\text{-}10}$ in $PM_{10}$) at **(b)** MSY and **(e)** MSA. Ångström-Asymmetry parameter matrix (scatterplot of AAE vs. *g* weighted by levels of $\%PM_{1\text{-}10}$ in $PM_{10}$) at **(c)** MSY and **(f)** MSA. (On hourly base).

**Figure 3.** Relationship between SSAAE and the relative contribution (%) of: **(a)** mineral dust to $PM_{10}$ at MSY and **(b)** $PM_{1\text{-}10}$ to $PM_{10}$ at MSA. Case studies discussed in the text show hourly SAE, AAE and SSAAE calculated for MSY during the periods **(c)** 28/06/2012 and **(d)** 15/10/2013-09/11/2013. Yellow and blue rectangles in Fig. 3d indicate the occurrence of SDE and precipitation respectively.

**Figure 4.** Scatterplot by bins between $OM_{ff}$ and HOA, and between $OM_{bb}$ and BBOA for $AAE_{bb}$=2 and $AAE_{ff}$=1 at MSY (on hourly base).

**Figure 5**. Scatterplot between $BC_{ff}$ and $NO_2$ for winter season (from November to February,) during the period 2012–2014 at MSY (daily base).

**Figure 6.** Daily cycle of: **(a)** AAE at MSY and MSA, **(b)** measured OM and simulated OM as the sum of $OM_{ff}$ and $OM_{bb}$ contributions at MSY, measured BC and simulated BC as the sum of $BC_{ff}$ and $BC_{bb}$ contributions at **(c)** MSY and **(d)** MSA. Annual cycle of: **(e)** AAE at MSY and MSA, **(f)** measured OM and simulated OM as the sum of $OM_{ff}$ and $OM_{bb}$ contributions at MSY, measured BC and simulated BC as the sum of $BC_{ff}$ and $BC_{bb}$ contributions at **(g)** MSY and **(h)** MSA. The study period ranges between 14/06/2012-09/07/2013 for OM contributions and between 12/06/2012-31/12/2014 for BC contributions, depending on the availability of BC and OM experimental measurements, respectively. Averages were calculated from hourly base.

**Figure 7.** Scatterplot by bins between AAE and $\%BC_{bb}$ at MSY and MSA. Error bars are one standard deviation of the averages calculated from daily values.

**Figure 8.** Daily cycle of: **(a)** AAE, **(b)** Measured OM and simulated OM as the sum of $OM_{ff}$ and $OM_{bb}$ contributions, **(c)** Measured BC and simulated BC as the sum of $BC_{ff}$ and $BC_{bb}$ contributions, during a wildfire episode (23 July 2012) at MSY.

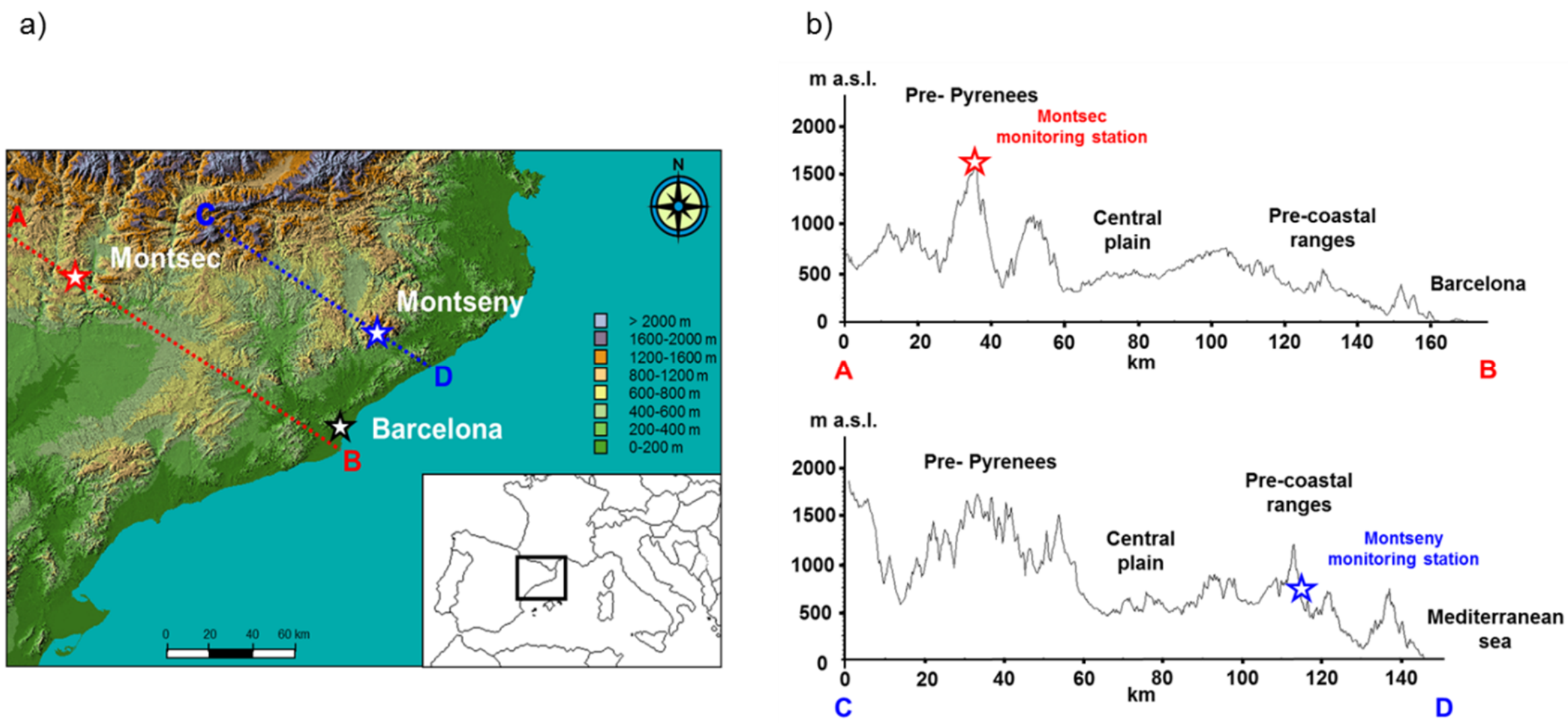

**Figure 1**

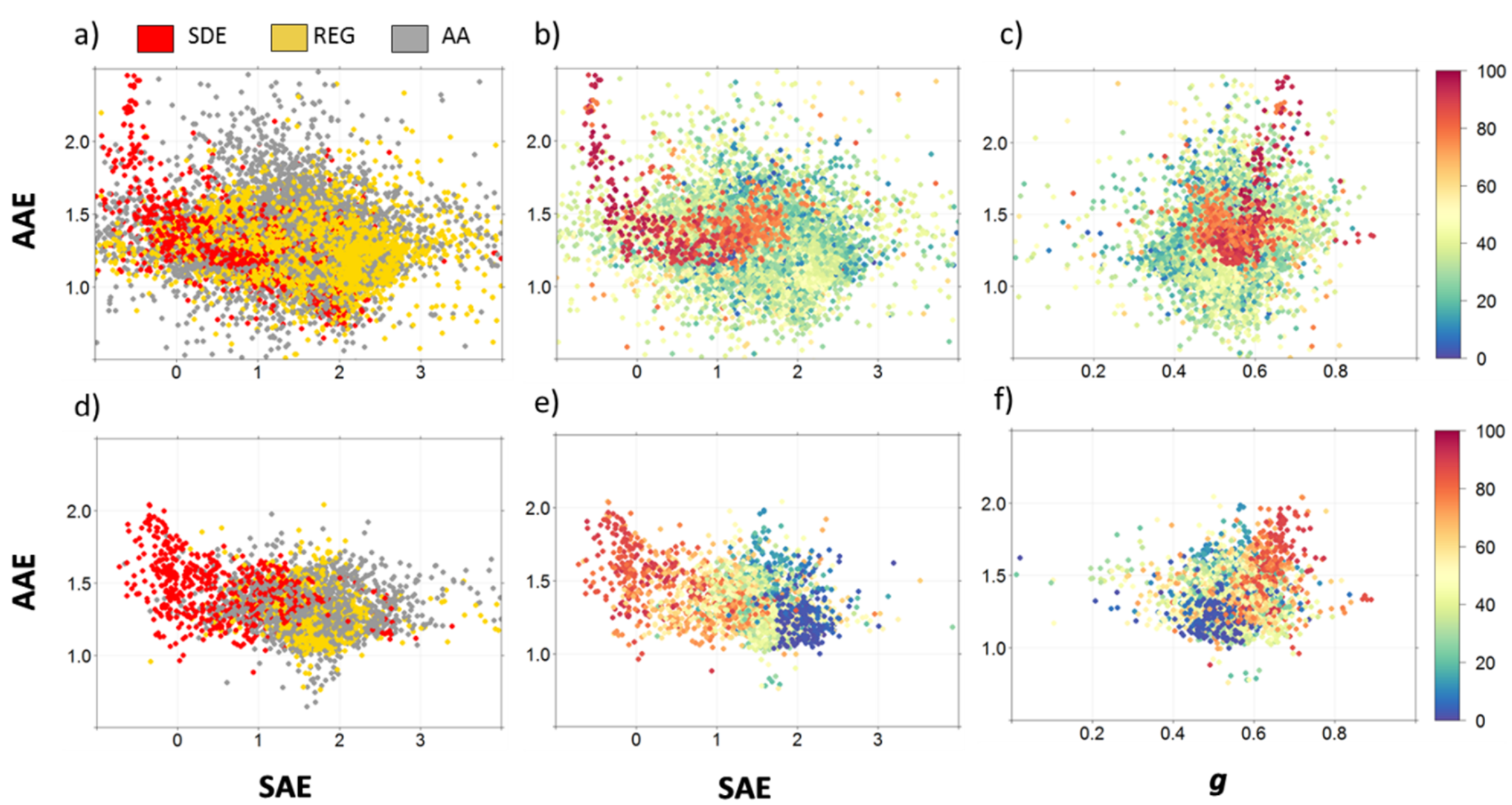

**Figure 2**

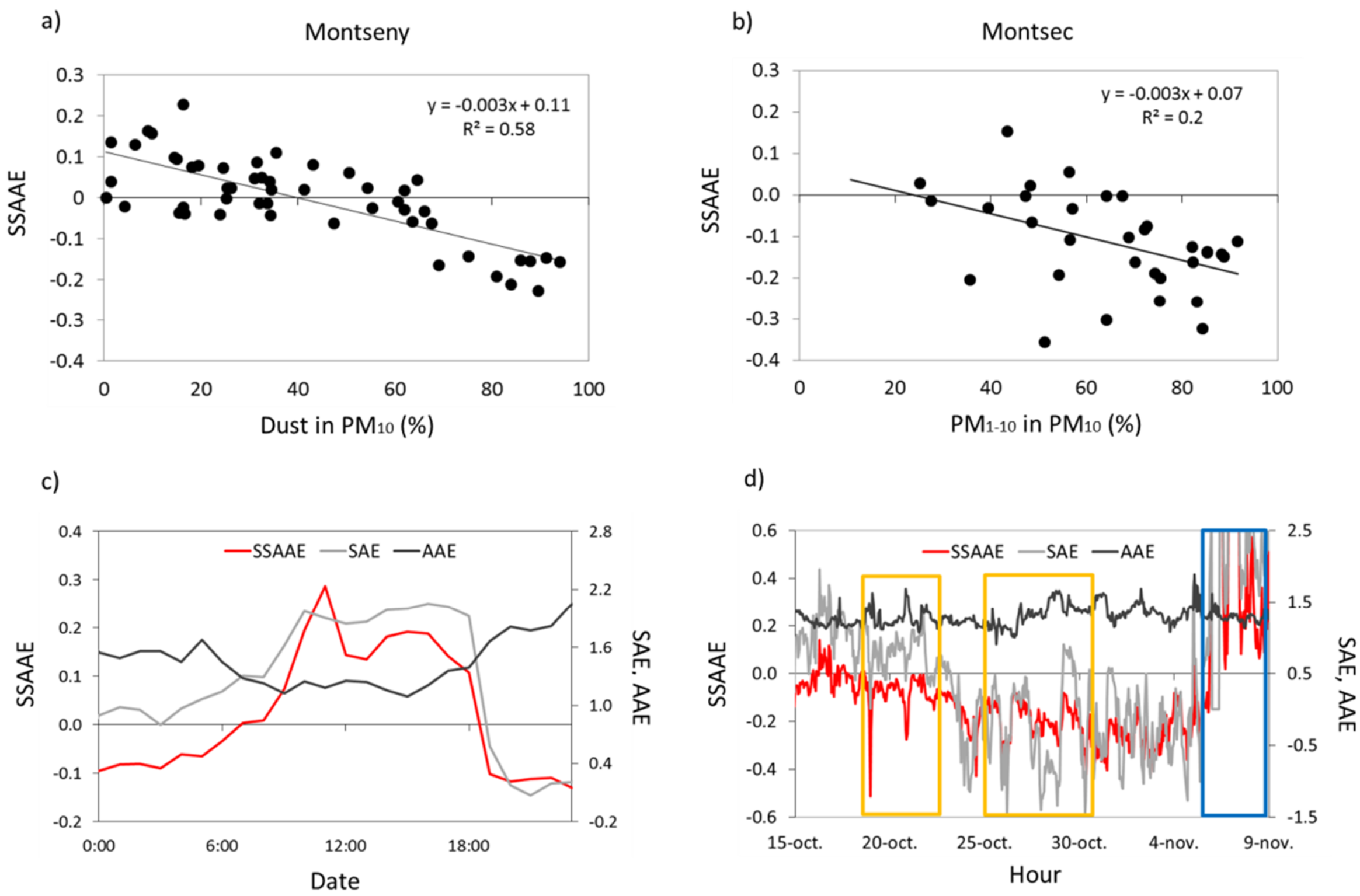

**Figure 3**

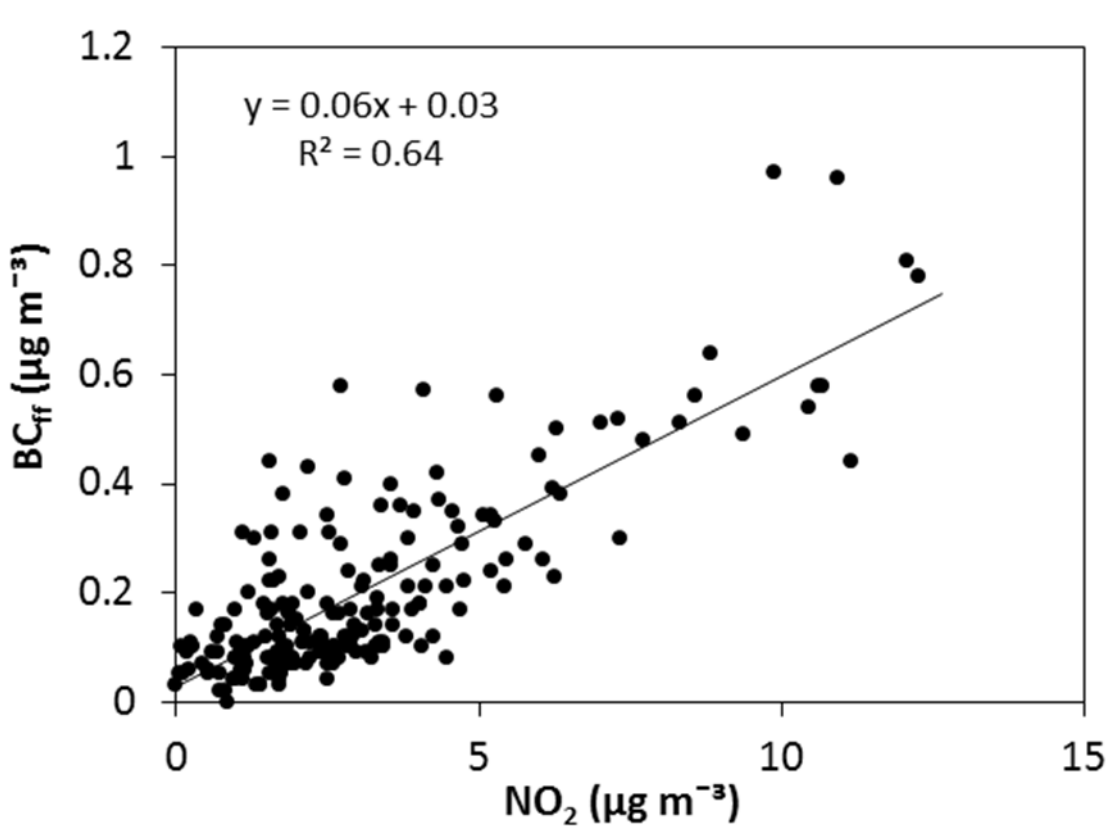

**Figure 4**

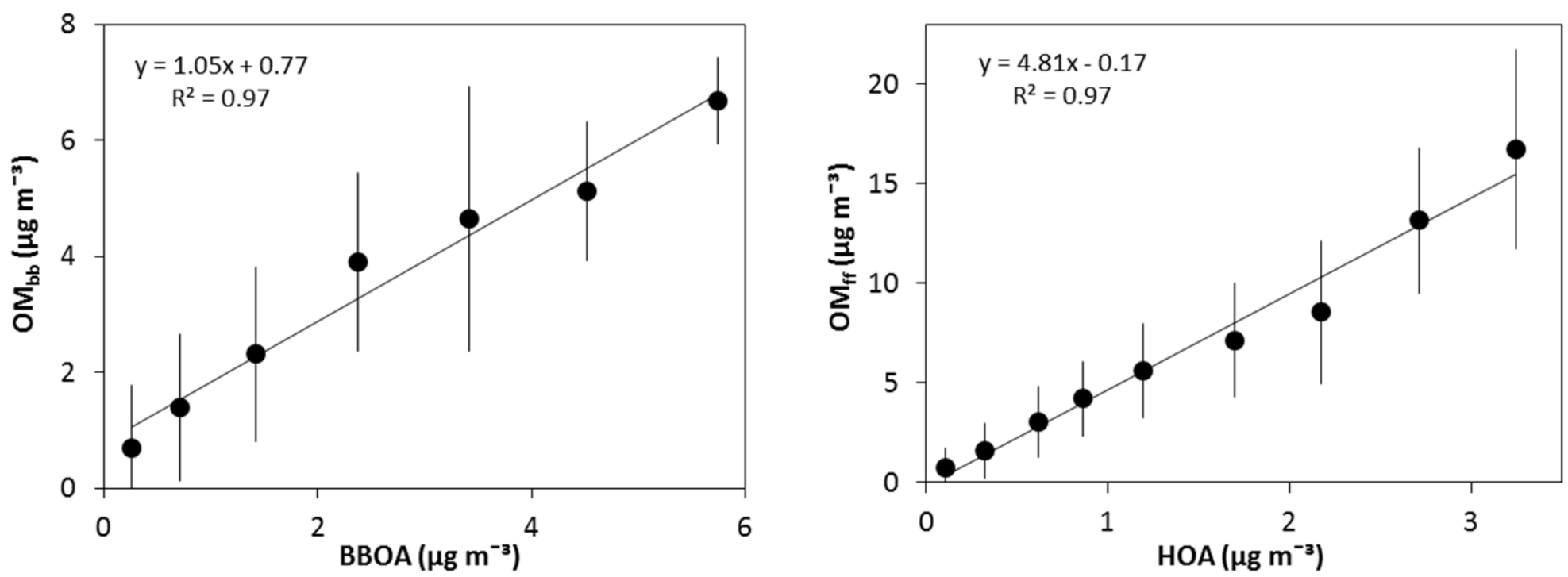

**Figure 5**

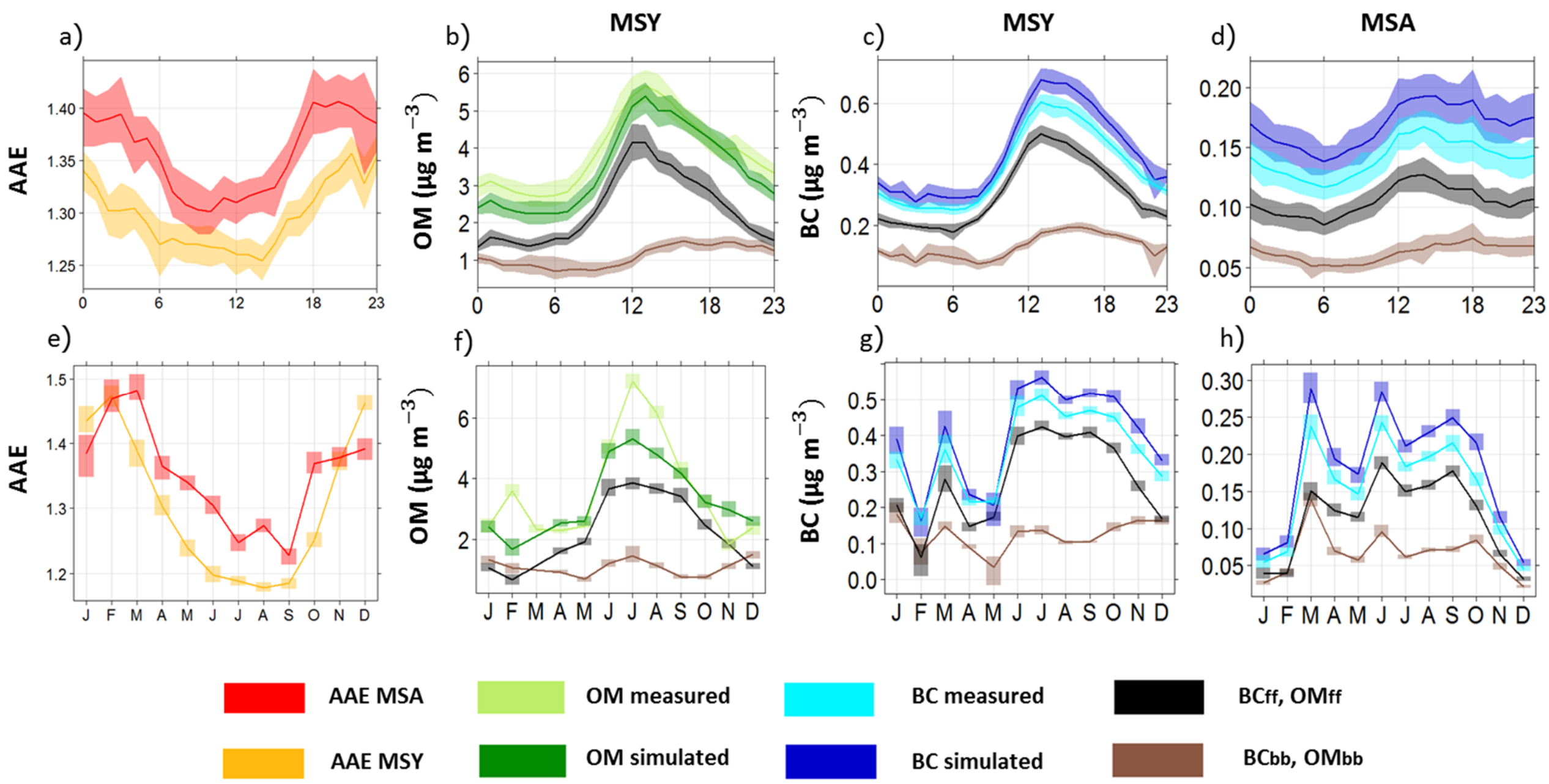

**Figure 6**

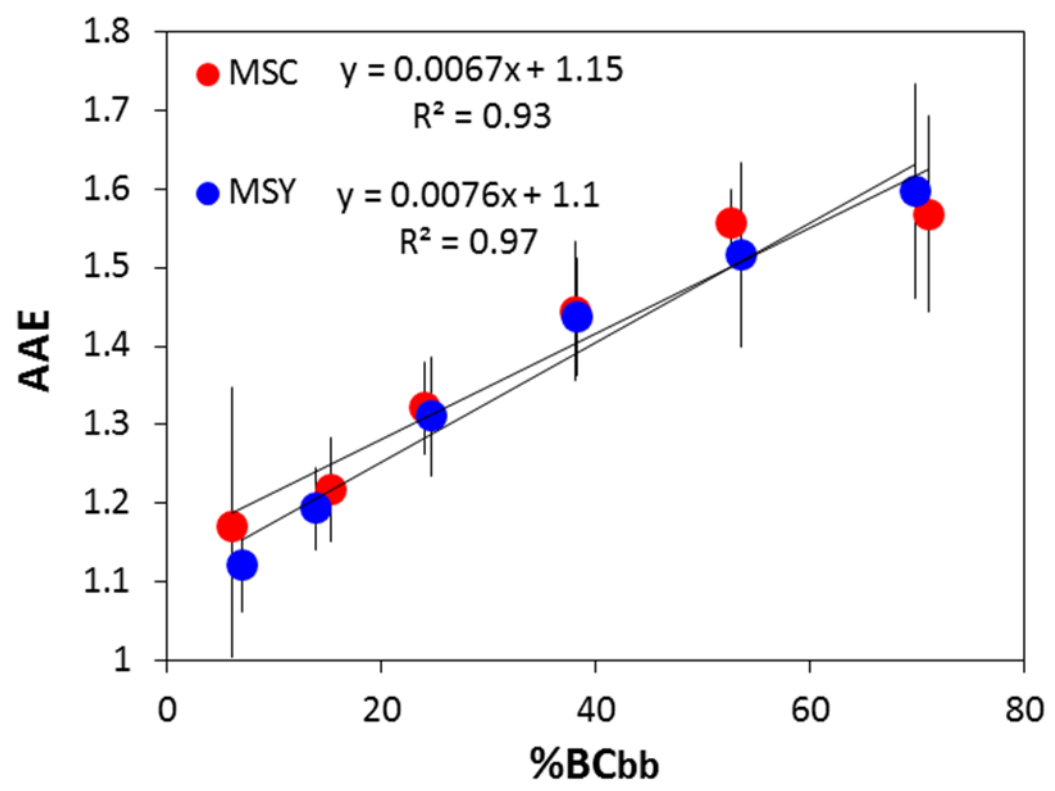

**Figure 7**

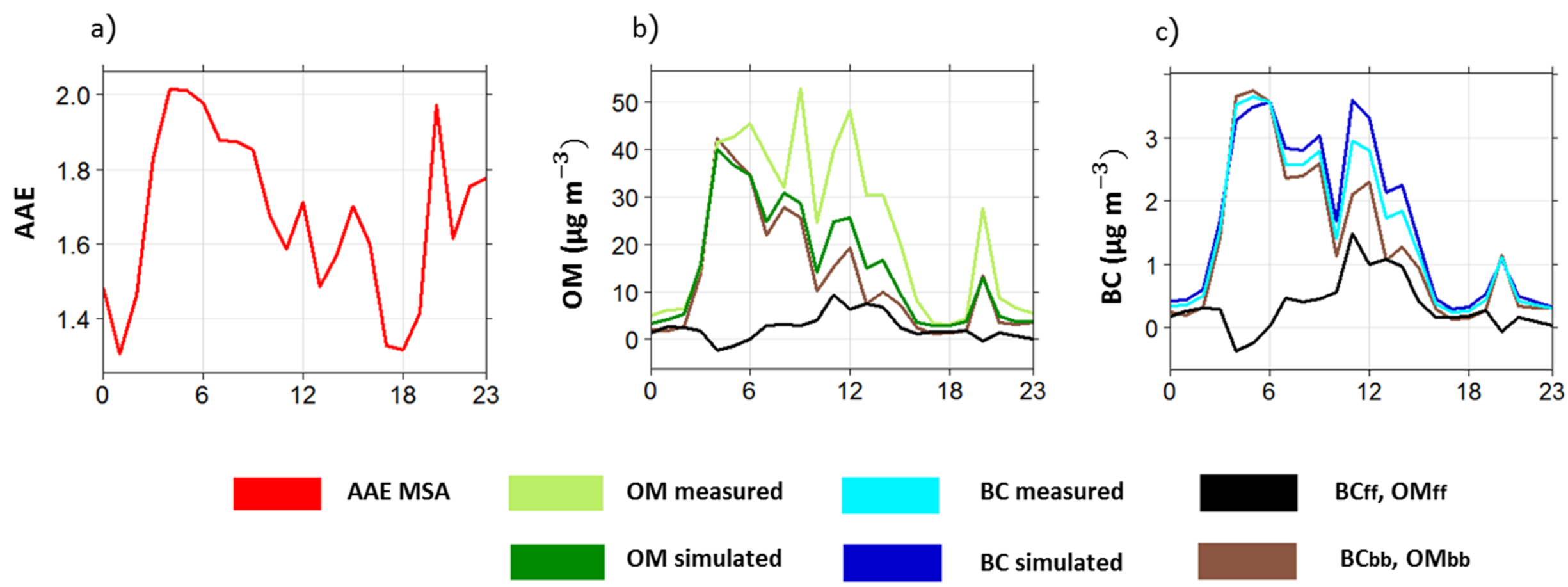

5 **Figure 8**