# Peer review of "Detection of Saharan dust and biomass burning events using near real-time intensive aerosol optical properties in the northwestern Mediterranean."

_Atmospheric Chemistry and Physics, 2015_

## Referee Comment (RC1) · Anonymous Referee #1 · 17 Feb 2016

General comments

1. This paper by Ealo et al. presents a very interesting idea for the real-time detection of dust and biomass burning events. However, one major concern I see with this technique is the difficulty to differentiate between the dust and biomass burning events, both dust and biomass being strong absorber in UV. This issue might be bigger in summer when the co-occurrence of SDE and Wildfire events may be highly probable. Due to re-circulation, these events may not be differentiated over prolonged time scales. 2. This technique make use of intrinsic properties of the aerosol species like Absorption, Scattering and Single Scattering Albedo Angstrom Exponents. However, these properties are influenced by environmental factors like temperature, RH, aerosol aging time,

etc., which is not discussed in this study. A part of difference in aerosol optical properties between MSY and MSA may be due to the fact that aerosol processing at these locations may be different and aerosol may have different properties. These concerns are highlighted especially during the re-circulation events. Please discuss.

Specific comments

1. The nephelometer instrument was calibrated only 3 times a year and zero adjust was carried out once a day may possibly insufficient for unbiased measurements. Some plots or data showing the stability of the instrument can be helpful in supporting the frequency of calibration and zero adjustments. 2. In order to help the reader, please provide average and standard deviation values in parentheses while comparing the optical properties in different events or between the two stations, 3. Line 511: "bellow" correction: below 4. Lines 592 and 593: please provide the abbreviated station names in the heading. 5. Lines 553- Lines 560: Thee fraction of BBOA and HOA in previous studies may be dependent upon the time of the year those measurements were made. So how fair it is to make those assumptions based on the observations in previous study? 6. Lines 651 and 678: "leaded" correction: lead 7. Supplementary tables S1 should be numbered S1 (a) and S1 (b) as they are discussed in the text.
* * *

---

## Author Comment (AC1) · 12 Apr 2016

acp-2015-902

**Detection of Saharan dust and biomass burning events using near real-time intensive aerosol optical properties in the northwestern Mediterranean.**

The authors would like to thank the reviewer for their comments and suggestions, which helped improving the quality of this work. A new version of the manuscript has been prepared following the suggestions. We provide below detailed replies to each of the comments.

**Anonymous referee #1**

Reviewer#1. General comment 1). **This paper by Ealo et al. presents a very interesting idea for the real-time detection of dust and biomass burning events. However, one major concern I see with this technique is the difficulty to differentiate between the dust and biomass burning events, both dust and biomass being strong absorber in UV. This issue might be bigger in summer when the co-occurrence of SDE and Wildfire events may be highly probable. Due to re-circulation, these events may not be differentiated over prolonged time scales. Please discuss.**

Reply to Reviewer#1. General comment 1). The authors agree with the referee that the concomitance of biomass burning and wildfires episodes during Saharan dust events (SDE) may be an issue. As stated in the manuscript, one of the objectives of this paper is to study the limitations of the proposed technique, firstly developed by Collaud Cohen et al. (2004), and to highlight the necessity of a better estimation of the different episodes through a multidisciplinary approach.

For example, as we show in the manuscript, wildfires events can be detected by means of different tools such as back-trajectories, forecast models and remote sensing data (satellite images, ceilometer/LIDAR and sun photometers measurements), and considered as isolated events (Fig. 8 in the manuscript). Concerning biomass burning events, the scattering Ångström exponent (SAE) parameter is certainly a useful parameter to establish differences between mineral dust (coarse material) and biomass burning (finer aerosol). Moreover, it is also possible to consider online PMx mass data in order to assess the SAE parameter regarding the changes in the size of aerosols. In addition, the black carbon concentration, which considerably increases during biomass burning events (See Fig. 8 describing a wildfire episode in the manuscript), can be also investigated. Furthermore, offline chemical filter analysis is a valuable input for the identification of iron oxides contained within the mineral dust to characterize the atmospheric situation.

It should be also considered that biomass burning is an active source in winter in the area under study and that during the study period (2012-2014) only three SDE were identified in winter. Furthermore, biomass burning concentration is relatively lower compared to northern areas in Europe; annual average contribution was estimated in 36%. During these winter SDE we did not see a marked daily cycle of AAE, therefore the dominance of mineral dust appears to be larger with respect to biomass burning regarding the effects on intensive optical properties.

In order to take into account the Referee's comment the following sentence has been added to section 5.3.3 in the revised manuscript:

"The concomitance of biomass burning and wildfire episodes during SDE may be an issue, being both dust and biomass burning strong absorbers in the UV. The SAE is a useful parameter that should be considered in order to establish differences in near real-time between mineral dust (coarse material) and biomass burning (finer aerosol). However, since relatively low biomass burning concentration was found in the area under study, the dominance of mineral dust appears to be larger with respect to biomass burning regarding the effects on intensive optical properties. Furthermore the co-occurrence of SDE and biomass burning winter emissions is not usual. Whereas for differentiating wildfires and SDE, both frequently occurring during summer, wildfires events can be considered as isolated events and detected by means of different tools such as back-trajectories, forecast models and remote sensing data".

**Reviewer#1. General comment 2). This technique makes use of intrinsic properties of the aerosol species like Absorption, Scattering and Single Scattering Albedo Angstrom Exponents. However, these properties are influenced by environmental factors like temperature, RH, aerosol aging time, which is not discussed in this study. A part of difference in aerosol optical properties between MSY and MSA may be due to the fact that aerosol processing at these locations may be different and aerosol may have different properties. These concerns are highlighted especially during the re-circulation events. Please discuss.**

Reply to Reviewer#1. General comment 2). The most important ambient parameter affecting scattering measurements is RH. High values of RH produce hygroscopic growth on particles, enhancing the scattering coefficient. It should be note that nephelometer measurements were made at constant RH, lower than 40%, following standards recommended by ACTRIS in order to avoid possible scattering enhancement due to hygroscopicity. Moreover, also the temperature (changing with RH) is controlled in the nephelometer cell and kept at almost constant values throughout the year.

Certainly, the aging of the aerosols affects the aerosol intensive optical properties (IOP). However, the study of aerosol aging is far beyond the scope of this work. In this manuscript we are mainly interested in studying how IOP change during SDE and biomass burning events, which dominate the IOP during their occurrence. Moreover, as we show in the manuscript, the proximity to anthropogenic sources of fine PM is the main affecting the SSAAE.

Despite this fact, the effect of aerosol aging time on the intensive optical properties is partially considered and discussed in the text within the air mass origin analysis, where Atlantic advections, Saharan dust events and re-circulation episodes were considered (Fig. 2a and 2d in the manuscript). Atlantic advections, occurring more frequently during winter, are characterized by dominance of fresh aerosols. MSA is usually within free troposphere during the colder period and influenced by clean air masses from Atlantic advections. However re-circulation episodes usually take place during warmer months, when both stations are frequently within the planetary boundary layer. These episodes are identified by stagnated air masses, high insolation and lack of precipitation lasting for few days, resulting in a dominance of aged aerosol. Probably the aerosol aging time is more pronounced at MSA, since it is more isolated from fresh anthropogenic pollutant sources transported mainly from Barcelona urban and industrial areas. Then pollutants at MSA present larger atmospheric residence time and thus also probably stronger aging process than at MSY.

Nevertheless, for a deeper evaluation of how these variables independently affect the intensive optical properties it would be necessary to perform specific studies, such as evaluating the aerosol aging time and coating processes.

**Reviewer#1. Specific comment 1). The nephelometer instrument was calibrated only 3 times a year and zero adjust was carried out once a day may possibly insufficient for unbiased measurements. Some plots or data showing the stability of the instrument can be helpful in supporting the frequency of calibration and zero adjustments.**

Reply to Reviewer#1. Specific comment 1). We calibrated both nephelometers following the standards proposed by ACTRIS network in order to obtain high quality data, and comparability among all ACTRIS stations. Both instruments have participated in workshop intercomparisons organized by ACTRIS and showed satisfactory comparability with other EU nephelometer and "reference" instruments.

As an example, Figure 1 shows the stability of nephelometer measurements at MSA station. Figure 1 shows the average scattering coefficients at the three wavelengths for different percentiles (0.05, 0.25, 0.50, 0.75, 0.95) during zero check calibration for the year 2013. Average ± standard deviation are reported for each wavelength during zero check. Since acceptable limits for zero check are set in ±2 Mm⁻1, our data are within the standards recommended by the manufacturer.

It should be noted that some problems concerning the stability of measurements were found after long-distance transportation. However, most of these problems were solved through a full calibration of the nephelometer.

[Figure]

**Figure 1.** Average scattering coefficients for the calculated percentiles (0.05, 0.50, 0.50, 0.75, 0.95) during zero check calibration measurements at 450, 525 and 635 nm for the year 2013 at MSA station. Average and standard deviation of scattering for the three wavelengths during the whole period are also reported.

**Reviewer#1. Specific comment 2). In order to help the reader, please provide average and standard deviation values in parentheses while comparing the optical properties in different events or between the two stations.**

Reply to Reviewer#1. Specific comment 2). We agree with the Referee that this information is missing in the manuscript. It should be considered, however, that data within the Ångström matrix are displayed on hourly base whereas air mass origin was identified once per day, therefore average values during a whole day could include some hourly data points which are not representing these situations. Nevertheless, ranges of SAE, AAE and %PM$_{1-10}$ for different air mass origin provided in the text are meaningful in order to consider limit values purely representing these atmospheric situations.

In order to consider the Referee's comment, the frequency distribution plots of SAE and AAE for SDE, regional episodes (REG) and Atlantic advections (ATL) will be added in the supporting material (Fig. 2a and 2b for MSY and MSA). These plots will help the reader to interpret the variability of the intensive properties and establish differences among atmospheric scenarios at both stations.

Accordingly to the referee's suggestion, the following sentence was added to section 5.2 of the Manuscript: "Further information providing the frequency distribution and average value of SAE and AAE for each atmospheric scenario is reported in Figs. S1a and S1b for MSY and MSA.

[Figure]

**Figure 2a.** Frequency distribution of SAE and AAE parameters for the three atmospheric scenarios (SDE REG, ATL) displayed in the Ångström matrix at MSY. Average and standard deviation of these parameters are also reported for the whole period (2012-2014).

[Figure]

**Figure 2b.** Frequency distribution of SAE and AAE parameters for the three atmospheric scenarios (SDE, REG, ATL) displayed in the Ångström matrix at MSA. Average and standard deviation of these parameters are also reported for the whole period (2013-2014).

**Reviewer#1. Specific comment 3). Line 511: "bellow" correction: below.**

Reply to Reviewer#1. Specific comment 3). It has been corrected.

**Reviewer#1. Specific comment 4). Lines 592 and 593: please provide the abbreviated station names in the heading.**

Reply to Reviewer#1. Specific comment 4). It has been corrected.

**Reviewer#1. Specific comment 5). Lines 553- Lines 560: Thee fraction of BBOA and HOA in previous studies may be dependent upon the time of the year those measurements were made. So how fair it is to make those assumptions based on the observations in previous study?**

Reply to Reviewer#1. Specific comment 5). Effectively, it is not far enough to consider the POA to SOA ratios from previous studies carried out during not identical periods of the year. Unfortunately, as stated in the manuscript, we did not performed any experiment, simultaneously to the study period, for differentiating primary from secondary organic sources (such as using levoglucosan as biomass burning tracer), and then we had to resort to results obtained in previous studies since it is the unique information we have to establish POA to SOA relations. In order to differentiate primary from secondary $OM_{bb}$, we considered the proportion of SOA originated from BBOA reported by Cubison et al. (2011). This ratio was primary applied to the results of the source apportionment to ACSM performed at MSY station (Minguillón et al., 2015), which were also used in this study. For $OM_{ff}$/HOA ratio, we considered results from previous studies performed also at MSY using $^{14}$C technique (Minguillón et al., 2009).

Thus, we do not have the necessary information to evaluate how much the use of non-simultaneous ancillary data is affecting our results. However, even though these assumptions concerning POA to SOA ratios are not from measurements carried out in the same study, the considered results for $AAE_{ff}$ and $AAE_{bb}$ (1 and 2 respectively) satisfactory agreed with those coefficients used in previous studies deploying the Aethalometer model.

Following the referee 's suggestion section 5.3.2 in the revised manuscript has been modified as follows:

"Since simultaneous experimental data to the study period were not deployed for differentiating POA to SOA ratios, we have considered results previously reported for MSY measurements. SOA formation from biomass burning emissions can be up to 25% of the BBOA emitted, as shown by Cubison et al. (2011) using Aerosol Mass Spectrometer (HR-ToF-AMS) data. This ratio was primarily applied by Minguillón et al. (2015) to the results obtained from the source apportionment to ACSM performed at MSY station, which were also used in this study."

**Reviewer#1. Specific comment 6). Lines 651 and 678: "leaded" correction: lead**

Reply to Reviewer#1. Specific comment 6). It has been corrected as "led" instead of "lead".

**Reviewer#1. Specific comment 7).** Supplementary tables S1 should be numbered S1 (a) and S1 (b) as they are discussed in the text.

Reply to Reviewer#1. Specific comment 7). It has been corrected.

---

## Referee Comment (RC2) · Anonymous Referee #3 · 27 May 2016

The present paper propose to investigate the potential of detection of Saharan dust and biomass burning events at surface stations. The proposed set-up is based on optical measurements (scattering and absorption) made with multi wavelength nephelometer and aethalometer. More specifically the idea is to use the wavelength dependence of scattering and absorption of the aerosol as a function of their composition (i.e of their sources). The topic of the paper is really very interesting since the possibility of distinguishing the aerosol source and composition is a big challenge for air quality control also in relation with climate change. Moreover it can participate to improve our skills to exploit the data available at supersites that have been set-up in the last decade over Europe in the framework of infrastructure programs like ACTRIS. For these

reasons, I think that this paper deserve to be published in ACP journal but some minor corrections and improvement are mandatory before publication. In general, i believe that a clear statement/summary about the possibility of detecting SDE and BB events is missing. What are the good conditions in which you can detect SDE event at the end? I have the impression that only almost "pure" SDE event are detectable and it is possible only at altitude/remote site. What is the best indicator to do that finally? I am not convince by SAE (a lot of overlap in scatter plots) and more by SSAAE. This is the same for BB events. What are at the end the uncertainties and what do you propose to improve this? is the present instrumentation enough or is it mandatory to have additional instruments such like ACSM ? I also regret that you never present the global amount of aerosol to have an idea of the relative importance of the dust or BB aerosol.

I now detail other remarks following the order of the text:

Abstract

Line 18 : here and elsewhere, you never clearly define the "Angstöm matrix".

Line 29 : at this stage FF has not been defined

This high number of acronym sometimes turn to madness, maybe it is useful to have somewhere the list of all acronym.

Introduction

Line 25 to 30 : I do not understand the logical link "as a consequence" between the two sentences and also I do understand the meaning of the following sentence "Given the huge . . .". I think that this part of the introduction as to be rephrase.

Line 12: Is Spain often having exceedances compare to other European countries ?

Chapter 3

P7 line 14 "assess"

**[ACPD](ACPD)**

Interactive
comment

P7 line 19: Maybe you need to tell more about the results of Russel et al (2010).

Finally, and it is link to one of general remark, you would say that the methodology for SDE only works for altitude (very remote sites) ?

P8 line 7 "in this work" which one ? Minguillon et al ? then do no go to the line.

Chapter 5

P9 line 2 to 8 there is something confusing in these lines, you compare SAE at MSA (altitude) and MSY (near Barcelona) and you say that SAE is greater at MSA because it is often within the BDL but probably not as often as MSY I guess. In line 4 you have to give some elements about the fact that we have smaller and darker particles at mountain sites. Line 6-7-8 I do not understand your analysis, I do not know what is compared to what.

P9 line 9 : do you have an analysis to propose about relative SSA values at each sites ?

P9 line 17 : maybe define the "angstrom matrix" term.

P9 line 18 : I think you never define what is PM1-10 , even if I can guess it is better to define it .

P9 line 21: I would have prefer "situation" than "scenario"

P9 line 22-23 : the limits you are mentioning are not that clear, it seems to me that there is an important mixing of the different "scenario" even if some patterns are indeed emerging. It would interesting to have the statistics of the nb of points within the limits of SAE and AAE for SDE and also for other situations. A more quantitative way to evaluate this aspect.

P10 line 5-15 once more you are speaking of the fact that we have finer particles at mountains top for example for AA situations (you definitely have to explain why I think) and on the other hand you are saying that we also have finer particles during REG due

to pollution. Is it meaning that we can not distinguish ? just PM concentrations help to decide this probably. You have to clarify the discussion and the objectives of the discussion

P10 line 7-8 what kind of aerosol concentrations are we speaking of?

P11 line 11-20 : the analysis of this episodes has to be reinforce. More than supposition, you must present facts of this recirculation event using meteorological information. You are speaking often of this case with strong conclusion then you have to give more evidence to readers.

P13 line 12 : also less VOC emissions during winter no ?

P13 line 4-7 : is there not a contradiction within this sentence ? I have the imprrssion you are saying one thing (low values of AAE during the day) and its contrary (increase of AAE during warmest hours of the day) . . .

P13 line 16 : "noted"

P15 line 3-5 : you suppose having less BB during night because of a thermal inversion above residual layer but is BB not also associated to domestic heating during colder hours i.e during the night which would imply that this BB would be trapped within the thin night BDL.

---

## Author Comment (AC2) · 21 Jul 2016

**Detection of Saharan dust and biomass burning events using near real-time intensive aerosol optical properties in the northwestern Mediterranean.**

The authors would like to thank the reviewer for their comments and suggestions, which helped improving the quality of this work. A new version of the manuscript has been prepared following the suggestions. We provide below detailed replies to each of the comments.

**Anonymous referee #3**

**Reviewer#3. General comment 1).** **The present paper propose to investigate the potential of detection of Saharan dust and biomass burning events at surface stations. The proposed set-up is based on optical measurements (scattering and absorption) made with multi wavelength nephelometer and aethalometer. More specifically the idea is to use the wavelength dependence of scattering and absorption of the aerosol as a function of their composition (i.e of their sources). The topic of the paper is really very interesting since the possibility of distinguishing the aerosol source and composition is a big challenge for air quality control also in relation with climate change. Moreover it can participate to improve our skills to exploit the data available at supersites that have been set-up in the last decade over Europe in the framework of infrastructure programs like ACTRIS. For these reasons, I think that this paper deserve to be published in ACP journal but some minor corrections and improvement are mandatory before publication.**

**In general, i believe that a clear statement/summary about the possibility of detecting SDE and BB events is missing. What are the good conditions in which you can detect SDE event at the end? I have the impression that only almost "pure" SDE event are detectable and it is possible only at altitude/remote site. What is the best indicator to do that finally? I am not convince by SAE (a lot of overlap in scatter plots) and more by SSAAE.**

Reply to Reviewer#3. General comment 1).

As we discussed throughout the article, detection of SDE by using the intensive optical properties are dependent on the degree of background pollution and the intensity of the event, since the effect of mineral dust on wavelength dependence may be hindered by anthropogenic pollutants. Thus, the intensive properties have to be calibrated for each sampling site, in order to better characterize the atmospheric aerosols.

Nevertheless, optical instruments, such as nephelometer and aethalometer, are widely used within monitoring networks such as ACTRIS and present several advantages. They are almost unattended, user-friendly, with relatively economic price, and provide near-real time measurements at high temporal resolution. Furthermore we have also shown (case study in the manuscript) that the intensive optical properties provided by these instruments give information which is more difficult to obtain by means of other instruments at high time resolution (i.e: optical particle counters), such as the identification of mineral dust in the atmosphere after SDE due to regional recirculation. Thus, ground base in-situ optical measurements are really useful to characterize the mineral dust aerosol at surface level and establish relations with air quality.

Despite this, we agree with the Reviewer that, depending on the background atmospheric conditions, not all SDE can be clearly detected using SSAAE or SAE. Presenting these limitations is also one of the aims of this work. In fact, so far, the SSAAE parameter was used for SDE identification at Jungfraujoch station, located at 3580 m.a.s.l., where the SDE detection efficiency was of 100% (Collaud Cohen et al., 2004). Thus, depending on the local conditions, it can be rather difficult to identify a "best indicator" for

SDE detection. The synergy between different available tools/instruments (i.e: forecast models, back trajectories analysis, and columnar measurements) is probably the best strategy for a better detection and characterization of these events.

In order to take into account the Referee's comment, the following sentence was added in the Conclusion section:

"Thus, depending on the background atmospheric conditions, not all SDE can be clearly detected using SAE, AAE and SSAAE parameters. And then additional information provided i.e. by forecast models, back trajectories analysis, and columnar measurements is also required in order to better detect and characterize these events. Nevertheless, aethalometer and nephelometer instruments provide near real time measurements and allow a fast detection of the impact of SDE at ground level. Furthermore, due to the sensitiveness for detecting changes in aerosol size and composition, SSAAE and Angstrom matrix tools are more sensitive compared to other near real-time measurements."

**Reviewer#3. General comment 2). This is the same for BB events. What are the end the uncertainities and what do you propose to improve this? Is the present instrumentation enough or is it mandatory to have additional instruments such like ACSM?**

Reply to Reviewer#3. General comment 2).

The aethalometer model technique is based on studying the wavelength dependence of aerosol absorption coefficient in order to characterize biomass burning emissions, being this compound a strong absorber in the UV. This technique provides satisfactory estimations of the temporal variability of the contributions for both emission sources, biomass burning and fossil fuel. However, the model presents some limitations for obtaining absolute concentrations. It should be kept in mind that filter sampling artefacts and hypothesis made for assessing OC-to-OM ratios are sources of uncertainty for the Aethalometer model, and these uncertainties are mostly related to the choice of the Ångström exponent.

A larger uncertainty was found for the apportionment of biomass burning to OM ($OM_{bb}$) compared to BC ($BC_{bb}$), since there are some organic compounds also absorbing in the UV, such as some aromatic compounds which originates from anthropogenic activities. This fact may lead to a slight overestimation of $OM_{bb}$ due to the overlapping in the UV absorption. Another issue presented in this model is related to the apportionment of biogenic compounds from non-combustion sources, specially in those places with large biogenic emissions and SOA formation, as occurs in our emplacement.

Consequently, it is recommended to carry out simultaneous measurements/experiments applying different techniques not based on optical methods in order to assess the quantification of biomass burning by the model. Some of these techniques have been applied in other studies, such as using levoglucosan as biomass burning tracer, or calibrating the model with the BBOA obtained from applying a source apportionment to ACSM/AMS data, as we did in our study. However, the calibration of the AAE coefficient does not resolve the challenge of discriminating the different compounds absorbing in the UV by optical methods, leading to an overestimation of $OM_{bb}$.

The differentiation of brown carbon originated from different emission sources by using optical measurements is a challenge, and therefore further analysis using additional techniques for determining biomass burning would be convenient to assess the Aethalometer model performances.

Nevertheless, the aethalometer is a very useful instrument which provides an advantageous technique for real-time air quality monitoring. And then further research in characterizing brown carbon by means of optical techniques is needed in order to exploit the possibilities of the instrument.

In order to consider the reviewer comment, the next sentence was added in the conclusion section of the revised manuscript:

"The differentiation of brown carbon originated from different emission sources by using optical measurements is a challenge, in particular the SOA formation and transformation processes. Due to the uncertainties presented by the aethalometer model for providing absolute concentrations, it is recommended to carry out simultaneous measurements/experiments applying different techniques not based on optical methods, such as using levoglucosan as BB tracer or calibrating the model with BBOA obtained from ACSM source apportionment, in order to assess the quantification of biomass burning by the model. Nevertheless, the aethalometer model is a very useful tool which provides satisfactory estimations of the temporal variability of the contributions for both, biomass burning and fossil fuel emission sources. And then further research in characterizing brown carbon by means of optical techniques is needed in order to exploit the possibilities of the instrument."

**Reviewer#3. General comment 3). I also regret that you never present the global amount of aerosol to have an idea of the relative importance of dust or BB aerosol.**

Reply to Reviewer#3. General comment 3).

We agree with Reviewer#3, PM concentrations will help to the reader for a better understanding of the article. This issue has been addressed in the specific comment 15.

Figure S1 has been added to the revised supporting material.

**Reviewer#3. Specific comment 1). Abstract. Line 18. Here and elsewhere, you never clearly define the "Angstrom matrix".**

Reply to Reviewer#3. Specific comment 1).

We agree with the Referee, this definition is missing in the manuscript. The next two sentences will be added to the revised manuscript:

In the abstract: "The Angström matrix (made up by SAE and AAE)"

The sentence was rephrased as follows in section 5.2: "The angstrom matrix is a useful tool to detect periods dominated by SDE (Russell et al. 2010), it consists of a scatter plot made up by SAE parameter in the x-axis and AAE parameter in the y-axis, providing information about aerosol size and composition, respectively."

**Reviewer#3. Specific comment 2). Line 29: at this stage FF has not been defined.**

Reply to Reviewer#3. Specific comment 2).

We agree with the Referee. The next definition has been added to the revised manuscript in the abstract section. "fossil fuel (FF)"

**Reviewer#3. Specific comment 3). This high number of acronym sometimes turn to madness, maybe it is useful to have somewhere the list of acronyms.**

Reply to Reviewer#3. Specific comment 3).

Excellent suggestion, a list of acronyms containing the most relevant definitions has been added to the supplement information document. The next sentence was added to the manuscript in the introduction section:

"A list of acronyms used in this work is provided in table S1."

Table S1: List of acronyms

| Acronym | Definition |
|---|---|
| AA | Atlantic advections |
| $AAE_{bb}$ | Fossil fuel absorption Ångström exponent |
| $AAE_{ff}$ | Biomass burning absorption Ångström exponent |
| BB | Biomass burning |
| BBOA | Biomass burning organic aerosol |
| BC | Equivalent black carbon |
| $BC_{ff}$ | Equivalent black carbon from fossil fuel source |
| $BC_{bb}$ | Equivalent black carbon from biomass burning source |
| FF | Fossil fuel |
| g | Asymmetry parameter |
| HOA | Hydrocarbon-like organic aerosol |
| MSA | Montsec |
| MSY | Montseny |
| OM | Organic matter |
| $OM_{bb}$ | Organic matter from biomass burning source |
| $OM_{ff}$ | Organic matter from fossil fuel source |
| PBL | Planetary boundary layer |
| REG | Regional atmospheric episodes |
| SAE | Scattering Ångström exponent |
| SDE | Saharan dust event |
| SOA | Secondary organic aerosol |
| SSA | Single scattering albedo |
| SSAAE | Single scattering albedo Ångström exponent |
| WMB | Western Mediterranean Basin |

**Reviewer#3. Specific comment 4). Introduction. Line 25 to 30: I do not understand the logical link "as a consequence" between the two sentences and also I do understand the meaning of the following sentence "Given the huge . . .". I think that this part of the introduction has to be rephrased.**

Reply to Reviewer#3. Specific comment 4).

The expression "as a consequence" was suppressed and the text has been edited as follows: "These intensive properties present a valuable input for climate models, which require accurate information concerning the variability of atmospheric composition for targeted species via comparison with observations (Laj et al., 2009)."

**Reviewer#3. Specific comment 5). Line 12: Is Spain often having exceedances compared to other European countries?.**

Reply to Reviewer#3. Specific comment 5)

In this paragraph we are highlighting the importance of SDE effects on air quality, and therefore the necessity of detecting SDE in near-real time. As is detailed in the text, southern European countries are frequently affected by SDE. In fact, in Spain dust outbreaks lead to the 70% of the exceedances in the $PM_{10}$ daily limit value (Escudero et al., 2007a) at most regional background sites in Spain. Then, Spain often present exceedances in the $PM_{10}$ daily limit value due to Saharan dust outbreaks, compared to other northern European countries which are distant from African dust sources, and then SDE does not present important effects on air quality.

**Reviewer#3. Specific comment 6). Chapter 3, P7 line 14 "assess".**

Reply to Reviewer#3. Specific comment 6)

It has been corrected.

**Reviewer#3. Specific comment 7). P7, line 19: Maybe you need to tell more about the results of Russell et al (2010).**

Reply to Reviewer#3. Specific comment 7)

What we found missing in this sentence, referring to the work presented by Russell et al. (2010), is to mention that data analyzed in that work was obtained from columnar measurements, whereas our work was performed at ground level. Accordingly to the Referee's suggestion, the sentence in section 3 has been modified as follows:

"Russell et al. (2010) has also performed the AAE and SSAAE parameters for full aerosol vertical columns obtained from sun-sky photometer retrievals, in order to characterize aerosol columns dominated by the two important sources of UV absorbing aerosols, biomass burning and Saharan dust."

**Reviewer#3. Specific comment 8). Finally, and it is link to one of the general remark, you would say that the methodology for SDE only works for altitude (very remote sites)?.**

Reply to Reviewer#3. Specific comment 8).

Answer to this question is further developed in the general comment #1.

 In this work we have shown that the intensive optical properties present some limitations for characterizing atmospheric aerosols, however these properties may be calibrated for each sampling site for a better performance of the results. The detection of SDE mainly depends on the emplacement background pollution and the intensity of the event, and then a larger mixture of mineral dust with anthropogenic pollutants will prevent a change in the spectral dependence of scattering and absorption. The smaller size of anthropogenic pollutants unified to the black colour provided by the combustion emission sources, turns into positive values the spectral dependence of the SSA.
Therefore a better detection of SDE by means of using the intensive optical properties will be performed in those sites less influenced by anthropogenic emission sources. However, the best approach is to combine all available tools in order to: first, predict the SDE by performing mineral dust and meteorological forecast models; second, detect the mineral dust layer before the plume reaches to the ground by performing remote data from lidar/ceilometer, sun-photometer and/or satellite products; and

finally, characterize the impact of mineral dust at ground level by the performance of ground measurements.

**Reviewer#3. Specific comment 9). P8 line 7 "in this work" which one? Minguillón et al? then do no go to the line.**

Reply to Reviewer#3. Specific comment 9).

"In this work" refers to the study that is being presented in this manuscript. In order to avoid confusions, the expression has been replaced by the next one: "In the present work".

**Reviewer#3. Specific comment 10). Chapter 5, P9, line 2 to 8. There is something confusing in these lines, you compare SAE at MSA (altitude) and MSY (near Barcelona) and you say that SAE is greater at MSA because it is often within the BDL but probably not as often as MSY I guess. In line 4 you have to give some elements about the fact that we have smaller and darker particles at mountain sites. Line 6-7-8 I do not understand your analysis, I do not know what is compared to what.**

Reply to Reviewer#3. Specific comment 10).

Line 2 to 8: Previous studies have shown that under low aerosol loadings at mountain top sites the aerosol mixture is preferentially composed of relatively smaller and darker particles (i.e Pandolfi et al., Andrews et al., and references therein). A larger SAE at MSA is probably due to the frequent position of the station within the free troposphere (FT) (above the PBL (planetary boundary layer)), predominantly in winter. However, MSY which is located at lower altitude and more frequently within the PBL, shows slightly lower SAE probably due to the influence of a more polluted background environment.
It may be possible that Referee#3 has misunderstood the sentence in the manuscript: "Mean SAE was higher at MSA station compared to MSY, which could be explained by a dominance of smaller particles on average at MSA likely due to frequent position of the station within the free troposphere in winter."

Line 4: Previous studies at MSA site have described the free troposphere conditions, occurring almost during the colder period and characterized by a marked reduced PM concentrations (Ripoll et al., 2014). A more detailed analysis relating FT conditions and optical properties at MSA was performed by Pandolfi et al. (2014a). This study showed that under very low $PM_1$ concentrations (<1.5 µg m$^{-3}$) at MSA, SSA and g parameters reached very low values around 0.84 and 0.43 respectively, whereas the SAE increased (Figure 6 in Pandolfi et al. 2014a). These low PM conditions at MSA were related to the prevalence of small particles with relatively higher absorption properties regardless of the considered atmospheric scenario. Low values of SSA at very low aerosol loading have been observed at other mountain top sites (i.e. Andrews et al. (2011) and references therein), and were related with an aerosol mixture in which large aerosol scattering particles have been preferentially removed (e.g. by cloud scavenging and/or deposition), leaving behind a relatively smaller and darker aerosol.

In order to take into account the referee's comment, the paragraph was rephrased as follows in section 5.1 of the revised manuscript:

"As already reported (Andrews et al., 2011; Berkowitz et al., 2011; Marcq et al., 2010; Pandolfi et al., 2014a), under low aerosol loadings at mountain top sites, in which large aerosols scattering particles have been preferentially removed, the aerosol mixture is mainly composed of relatively smaller and darker particles. Previous studies at MSA have described the free troposphere conditions, characterized by very low $PM_1$ concentrations (<1.5 µg m$^{-3}$), low values of SSA (0.83) and g (0.43) parameter, and increasing SAE (Pandolfi et al., 2014a)."

Line 6-7-8: In order to avoid confusion, the sentence was rephrased as follows:
"MSY site presented slightly lower AAE values compared to MSA, due to a major predominance of black carbon particles as a consequence of the proximity to Barcelona urban area."

**Reviewer#3. Specific comment 11). P9 line 9: do you have an analysis to propose about relative SSA values at each sites?**

Reply to Reviewer#3. Specific comment 11).

Previous works focused on studying optical properties at MSY and MSA were performed by Pandolfi et al. (2011 and 2014a, respectively), and SSA values were analysed in these studies. For this reason, and also due to the fact that the intensive optical properties presented in our work aimed to detect specific atmospheric scenarios (SDE and BB events), we have not considered further analysis on SSA.

**Reviewer#3. Specific comment 12). P9 line 17: maybe define the "angstrom matrix" term.**

Reply to Reviewer#3. Specific comment 12).

We agree with the Referee. The text has been edited as follows:

"The angstrom matrix is a useful tool to detect periods dominated by SDE (Russell et al., 2010), it consist of a scatterplot made up by SAE parameter in the x axis and AAE parameter in the y axis, providing information about aerosol size and composition, respectively. The scatterplot can be colour coded and investigated by other parameters in order to further characterize the atmospheric aerosols. In our case the matrix was colour coded by different air mass origin and by the coarse fraction contained within the $PM_{10}$ (%$PM_{1-10}$ in $PM_{10}$), which was calculated as the difference between %$PM_1$ and %$PM_{10}$ contained within the $PM_{10}$ fraction."

**Reviewer#3. Specific comment 13). P9 line 18 : I think you never define what is PM1-10 , even if I can guess it is better to define it .**

Reply to Reviewer#3. Specific comment 13).

Following the referee's suggestion, the revised manuscript has been modified as is detailed in the previous answer:

"In our case the matrix was colour coded by different air mass origin and by the coarse fraction contained within in the $PM_{10}$ (%$PM_{1-10}$ in $PM_{10}$), which was calculated as the difference between %$PM_1$ and %$PM_{10}$ contained within the $PM_{10}$ fraction."

**Reviewer#3. Specific comment 14). P9 line 21: I would have prefer "situation" than "scenario"**

Reply to Reviewer#3. Specific comment 14).

It has been changed; "scenario" was replaced by "situation".

**Reviewer#3. Specific comment 15). P9 line 22-23: the limits you are mentioning are not that clear, it seems to me that there is an important mixing of the different "scenario" even if some patterns are indeed emerging. It would interesting to have the statistics of the nb of points within the limits of SAE and AAE for SDE and also for other situations. A more quantitative way to evaluate this aspect.**

Reply to Reviewer#3. Specific comment 15).

We agree with Referee#3, a quantitative analysis of SAE and AAE parameters for the different atmospheric situations should be added to the manuscript for a better performance of the intensive optical properties. However, it should be considered that data within the Ångström matrix is displayed on hourly base whereas air mass origin was identified once per day, therefore daily average values may include some hourly data points which are not representing entirely these situations.

In order to consider this comment, the general comment #3 and the specific comment #2 from Referee#2 regarding this issue, figure S1 has been added to the revised supporting material. These figures show the frequency distribution of SAE, AAE, $PM_{10}$ (µg m$^{-3}$) and %$PM_{1-10}$ in $PM_{10}$ for the different atmospheric scenarios (SDE, REG and AA), at MSY and MSA respectively. These plots will help to the reader to interpret the variability of the intensive properties and establish differences among atmospheric scenarios at both stations. In addition, the following sentences were added to section 5.2 in the revised manuscript:

"Average and standard deviation of SAE and AAE during SDE were 1.12±0.87 and 1.27±0.24 for MSY, and 0.69±0.78 and 1.41±0.25 for MSA. Lower SAE and higher AAE at MSA pointed to a larger dominance of mineral dust and a purer composition during these events at the high altitude station (Fig. S1). Average $PM_{10}$ concentrations during SDE were 25.4±17 and 21.0±17 µg m$^{-3}$ for MSY and MSA respectively. Further information providing the frequency distribution and average values of SAE, AAE, $PM_{10}$ and %$PM_{1-10}$ in $PM_{10}$ for each atmospheric situation at both stations is reported in Fig. S1."

"Average and standard deviation of SAE and AAE during these scenarios were 1.35±0.95 and 1.33±0.27 for MSY, and 1.65±0.57 and 1.30±0.16 for MSA, respectively. $PM_{10}$ showed the lowest concentrations during these events, being 11±7 and 9.4±6 µg m$^{-3}$ respectively for MSY and MSA (Fig. S1)."

"SAE and AAE values during REG episodes were 1.61±0.87 and 1.24±0.19 for MSY, and 1.66±0.48 and 1.29±0.15 for MSA, respectively. The average $PM_{10}$ concentrations during these atmospheric situations were 15.6±8 and 12.6±7 µg m-3.for MSY and MSA (Fig. S1)."

[Figure]

Figure S1. Frequency distribution, average and standard deviation of SAE, AAE, $PM_{10}$ and $\%PM_{1-10}$ in $PM_{10}$ parameters for the three atmospheric scenarios (SDE, REG, AA) displayed in the Ångström matrix at (a) MSY and (b) MSA.

**Reviewer#3. Specific comment 16). P10 line 5-15 once more you are speaking of the fact that we have finer particles at mountains top for example for AA situations (you definitively have to explain why I think) and on the other hand you are saying that we also have finer particles during REG due to pollution. Is it meaning that we cannot distinguish? just PM concentrations help to decide this probably. You have to clarify the discussion and the objectives of the discussion.**

Reply to Reviewer#3. Specific comment 16).

Definitely, Table S1 added to the supporting material is necessary for a better understanding of the intensive optical properties variation within the different atmospheric situations.

As we have discussed in specific comment #10, the high altitude MSA station is frequently located within the free troposphere during the colder period. The low temperature during this period restrict the atmospheric convection processes leading to a less growing of the PBL, and then, as a result the PBL height is lower than the altitude of the station (1570 m a.s.l.), which remains within the free troposphere. Free troposphere conditions at MSA can occur during both, AA and REG atmospheric situations taking place during winter, although AA (65%) scenarios are most common in this time of the year compared to REG (10%) (Ripoll et al., 2014).
Therefore, SAE and AAE parameters are very similar at MSA and only can be differentiated by $PM_{10}$ concentration (Fig. S1). $PM_{10}$ tail distribution is slightly shifted towards larger concentrations during REG compared to AA scenarios. However differences at MSY between both atmospheric situations are more evident, SAE at MSY present larger values during REG compared to AA episodes. It is also reflected in the $PM_{10}$ distribution which presents larger values during REG.

Following these results, we totally agree with the Referee#3 that $PM_{10}$ concentration provide valuable information and should be considered to establish differences among different atmospheric scenarios. Despite SAE values for AA and REG episodes are quite similar; the $PM_{10}$ concentrations mark the difference. REG episodes are related to pollution scenarios and show a higher average $PM_{10}$ concentration (12.6 and 15.6 µg m$^{-3}$ at MSA and MSY), whereas AA clears out the atmosphere leading to lower background concentrations (9.4 and 11.0 µg m$^{-3}$ at MSA and MSY).

**Reviewer#3. Specific comment 17). P10 line 7-8 what kind of aerosol concentrations are we speaking of?**

Reply to Reviewer#3. Specific comment 17).

Regarding to specific comments #16 and #17, the table S1 has been added to the revised supporting material. Frequency distribution and average values of SAE, AAE, $PM_{10}$ and $\%PM_{1-10}$ in $PM_{10}$ are provided for the three atmospheric scenarios (SDE, REG, AA) displayed in the Ångström matrix. Average $PM_{10}$ concentrations during AA scenarios were 11.0 and 9.4 µg m$^{-3}$ at MSY and MSA sites, respectively.

**Reviewer#3. Specific comment 18). P11 line 11-20 : the analysis of this episodes has to be reinforce. More than supposition, you must present facts of this recirculation event using meteorological information. You are speaking often of this case with strong conclusion then you have to give more evidence to readers.**

Reply to Reviewer#3. Specific comment 18).

In order to better characterize the regional recirculation scenario, the dust forecast image from Dream model was added to Fig. S3 in the revised supporting material (Fig. S3d).

The SDE is clearly identified by the air mass origin from North Africa (Fig S3a) and is further confirmed by the Dream model pointing to 20-40 µg m$^{-3}$ of dust surface concentration (Fig. S3c). The air mass origin during REG episode on 1/11/2013, occurring after the SDE, shows the stagnation of air masses with a very local and regional origin (Fig S3b). However, despite the intensive optical parameters show the presence of mineral dust during the REG episode with SSAAE<0 and the corresponding decreasing SAE and increasing AAE (Fig. 3d in the manuscript), the Dream model is not able to predict the resuspension of mineral dust (Fig. S3d).

PM$_{10}$ data is not available for this case study in order to evaluate PM$_{10}$ concentrations during the SDE and REG episode. We cannot further characterize the REG episode with the available observational data.

It would be interesting to carry out a simulation at high spatial resolution in order to better characterize the regional air masses leading the re-circulation of mineral dust. In fact, it would be interesting to deploy a study focused on the characterization of these re-circulation scenarios in order to better evaluate the effects on air quality, but we did not consider implementing high computing time simulations in this study.

[Figure]

**Figure S3.** Backward trajectories corresponding to (a) the Saharan dust event (28/10/2013) and (b) the regional episode (01/11/2013) atmospheric scenarios at MSY. Dust surface concentration at MSY from the Dream model corresponding to (c) the Saharan dust event (28/10/2013) and (d) the regional episode (01/11/2013).

Accordingly to referee's suggestion the next text was added to section 5.2 of the manuscript:

"It is interesting to highlight that despite the intensive optical parameters showed the presence of mineral dust during the REG episode, with SSAAE<0 and the corresponding decreasing SAE and increasing AAE, the Dream was not able to reproduce the recirculation of mineral dust (Fig. S3d), and only a simulation at high spatial resolution could characterize the event."

**Reviewer#3. Specific comment 19). P13 line 12 : also less VOC emissions during winter no ?**

Reply to Reviewer#3. Specific comment 19).

We agree with the Referee, the lower SOA formation in winter is mainly leaded by a less concentration of VOCs, which are one of the primary precursor sources of SOA formation during the warmer period in this emplacement (Seco et al. 2013).
The sentence will be rephrased as follows in the revised manuscript:

"The slope was close to the unity due: to the lower SOA formation in winter, mainly explained by a decreasing of VOCs emissions being one of the primarily precursor sources of SOA formation during the warmer period in this emplacement (Seco et al., 2013), and also as consequence of less photochemistry activity and the prevalence of primary emissions."

**Reviewer#3. Specific comment 20). P13 line 4-7 : is there not a contradiction within this sentence ? I have the impression you are saying one thing (low values of AAE during the day) and its contrary (increase of AAE during warmest hours of the day) . . .**

Reply to Reviewer#3. Specific comment 20).

We agree with the referee, there is a mistake in this sentence which leads to a contradiction regarding the daily cycle of AAE. The sentence was rephrased as follows:

"Low values of AAE during the day and higher at night at both sites resulted mainly from the development of sea and mountain breezes, favouring the transport of anthropogenic pollutants from the urbanized/industrialized coastline and valleys to inland areas and leading to a decrease of AAE during the warmest hours of the day (Fig. 6a).

**Reviewer#3. Specific comment 21). P14 line 16 : "noted"**

Reply to Reviewer#3. Specific comment 21). It has been corrected

**Reviewer#3. Specific comment 22). P15 line 3-5: you suppose having less BB during night because of a thermal inversion above residual layer but is BB not also associated to domestic heating during colder hours i.e during the night which would imply that this BB would be trapped within the thin night BDL.**

Reply to Reviewer#3. Specific comment 22).

We did not suppose having less BB during night. What we suppose in these lines is that $OM_{bb}$, which started increasing during the afternoon, remained stable during night and did not showed much variation (but the concentration was still high during night). This fact may be explained by the decrease of the PBL and the formation of the typical thermal inversion at lower levels during night, which may imply that the biomass burning emitted during the previous hours was trapped above the thermal inversion in the residual layer.

We agree with the Referee that BB concentration should increase within the night boundary layer as a result of domestic heating activities. However, since the station is located at higher altitude than emission sources (houses), we suppose that the BB we are measuring at night is trapped within the residual layer.

[revised manuscript text omitted]

5    **Figure 2**

[Figure]

**Figure 3**

[Figure]

5    **Figure 4**

[Figure]

**Figure 5**

[Figure]

5   **Figure 6**

[Figure]

**Figure 7**

[Figure]

**Figure 8**

**Supplementary material of "Detection of Saharan dust and biomass burning events using near real-time intensive aerosol optical properties in the northwestern Mediterranean"**

**Table S1.** List of acronyms

| Acronym | Definition |
| --- | --- |
| AA | Atlantic advections |
| $AAE_{bb}$ | Fossil fuel absorption Ångström exponent |
| $AAE_{ff}$ | Biomass burning absorption Ångström exponent |
| BB | Biomass burning |
| BBOA | Biomass burning organic aerosol |
| BC | Equivalent black carbon |
| $BC_{ff}$ | Equivalent black carbon from fossil fuel source |
| $BC_{bb}$ | Equivalent black carbon from biomass burning source |
| FF | Fossil fuel |
| g | Asymmetry parameter |
| HOA | Hydrocarbon-like organic aerosol |
| MSA | Montsec |
| MSY | Montseny |
| OM | Organic matter |
| $OM_{bb}$ | Organic matter from biomass burning source |
| $OM_{ff}$ | Organic matter from fossil fuel source |
| PBL | Planetary boundary layer |
| REG | Regional atmospheric episodes |
| SAE | Scattering Ångström exponent |
| SDE | Saharan dust event |
| SOA | Secondary organic aerosol |
| SSA | Single scattering albedo |
| SSAAE | Single scattering albedo Ångström exponent |
| WMB | Western Mediterranean Basin |

**Table S2~1~.** Statistics from the hourly averages of the considered aerosol parameters for the period under study at MSY (~a~above) and MSA (below~b~) sites.

| MSY | λ | Counts | Mean | SD | Median | Min | Max | Skewness | | | Percentiles | | |
|---|---|---|---|---|---|---|---|---|---|---|---|---|---|
| | | | | | | | | | 5 | 25 | 50 | 75 | 95 |
| $\sigma_{sp}$ | 635 | 28443 | 30.05 | 27.67 | 23.53 | -0.50 | 596.55 | 3.12 | 3.01 | 11.64 | 23.53 | 40.06 | 79.46 |
| | 525 | 28522 | 38.41 | 34.03 | 30.26 | -0.50 | 539.71 | 2.52 | 4.27 | 14.98 | 30.26 | 51.72 | 100.42 |
| | 450 | 28540 | 47.26 | 41.29 | 37.35 | -0.49 | 513.67 | 2.36 | 5.14 | 18.29 | 37.35 | 64.34 | 122.67 |
| $\sigma_{bsp}$ | 635 | 25894 | 4.21 | 3.38 | 3.63 | -0.50 | 60.56 | 2.12 | 0.26 | 1.77 | 3.63 | 5.87 | 10.06 |
| | 525 | 25974 | 4.67 | 3.76 | 4.01 | -0.50 | 107.08 | 2.82 | 0.40 | 1.99 | 4.01 | 6.47 | 11.14 |
| | 450 | 25951 | 5.46 | 4.23 | 4.70 | -0.50 | 58.86 | 2.08 | 0.57 | 2.40 | 4.70 | 7.51 | 12.86 |
| g | 635 | 23963 | 0.54 | 0.10 | 0.54 | -0.97 | 0.97 | -3.34 | 0.40 | 0.50 | 0.54 | 0.59 | 0.67 |
| | 525 | 24503 | 0.59 | 0.06 | 0.59 | -0.46 | 0.90 | -1.39 | 0.49 | 0.56 | 0.59 | 0.62 | 0.68 |
| | 450 | 25038 | 0.60 | 0.07 | 0.61 | -0.94 | 0.88 | -2.30 | 0.49 | 0.58 | 0.61 | 0.64 | 0.68 |
| $\sigma_{ap}$ | 470 | 21580 | 7.66 | 6.50 | 6.04 | -0.25 | 94.05 | 2.48 | 1.06 | 3.12 | 6.04 | 10.35 | 19.37 |
| | 880 | 21567 | 3.51 | 2.99 | 2.73 | -0.21 | 31.43 | 2.10 | 0.44 | 1.37 | 2.73 | 4.81 | 9.02 |
| SAE | 450-635 | 27959 | 1.38 | 0.79 | 1.42 | -2.45 | 5.98 | 0.14 | 0.06 | 1.01 | 1.42 | 1.76 | 2.48 |
| AAE | 370-950 | 21390 | 1.30 | 0.30 | 1.27 | -1.86 | 5.84 | 0.69 | 0.91 | 1.14 | 1.27 | 1.44 | 1.75 |
| SSA | 470 | 13585 | 0.83 | 0.07 | 0.84 | 0.12 | 0.98 | -1.69 | 0.70 | 0.80 | 0.84 | 0.88 | 0.91 |
| | 880 | 13575 | 0.80 | 0.12 | 0.83 | 0.05 | 1.00 | -1.71 | 0.57 | 0.75 | 0.83 | 0.89 | 0.95 |
| $PM_{10}$ | - | 35354 | 16.23 | 11.08 | 14.15 | 0.15 | 236.51 | 2.24 | 3.49 | 8.14 | 14.15 | 22.01 | 35.22 |

| MSA | λ | Counts | Mean | SD | Median | Min | Max | Skewness | | Percentiles | | | | |
|---|---|---|---|---|---|---|---|---|---|---|---|---|---|---|
| | | | | | | | | | | 5 | 25 | 50 | 75 | 95 |
| | 635 | 21708 | 16.73 | 19.28 | 9.37 | -0.50 | 307.33 | 2.05 | 0.22 | 2.77 | 9.37 | 25.19 | 54.79 | |
| $\sigma_{sp}$ | 525 | 21790 | 22.12 | 25.09 | 12.44 | -0.50 | 277.46 | 1.85 | 0.40 | 3.54 | 12.44 | 33.46 | 71.78 | |
| | 450 | 21792 | 28.11 | 31.50 | 16.06 | -0.50 | 376.38 | 1.80 | 0.59 | 4.53 | 16.06 | 42.70 | 91.38 | |
| | 635 | 21728 | 2.29 | 2.51 | 1.38 | -0.50 | 30.59 | 1.52 | -0.13 | 0.35 | 1.38 | 3.68 | 7.14 | |
| $\sigma_{bsp}$ | 525 | 21757 | 2.69 | 2.91 | 1.60 | -0.50 | 36.04 | 1.45 | -0.09 | 0.41 | 1.60 | 4.30 | 8.36 | |
| | 450 | 21542 | 3.19 | 3.43 | 1.94 | -0.50 | 42.44 | 1.42 | -0.10 | 0.46 | 1.94 | 5.16 | 9.82 | |
| | 635 | 18287 | 0.52 | 0.17 | 0.54 | -1.00 | 0.94 | -3.10 | 0.25 | 0.48 | 0.54 | 0.60 | 0.71 | |
| g | 525 | 18657 | 0.57 | 0.14 | 0.59 | -1.00 | 0.94 | -3.53 | 0.35 | 0.54 | 0.59 | 0.63 | 0.73 | |
| | 450 | 18644 | 0.60 | 0.14 | 0.62 | -1.00 | 0.94 | -3.63 | 0.38 | 0.57 | 0.62 | 0.66 | 0.77 | |
| $\sigma_{ap}$ | 470 | 9913 | 3.57 | 3.95 | 2.03 | -0.24 | 70.52 | 2.53 | 0.13 | 0.65 | 2.03 | 5.64 | 10.50 | |
| | 880 | 9915 | 1.59 | 1.71 | 0.89 | -0.16 | 23.03 | 1.79 | 0.06 | 0.29 | 0.89 | 2.57 | 4.81 | |
| SAE | 450-635 | 20189 | 1.58 | 0.83 | 1.64 | -1.95 | 6.00 | -0.04 | 0.08 | 1.24 | 1.64 | 1.95 | 2.82 | |
| AAE | 370-950 | 9625 | 1.36 | 0.27 | 1.32 | -0.90 | 4.80 | 2.02 | 1.05 | 1.21 | 1.32 | 1.47 | 1.76 | |
| SSA | 470 | 7146 | 0.85 | 0.08 | 0.87 | 0.21 | 1.00 | -2.16 | 0.69 | 0.82 | 0.87 | 0.90 | 0.93 | |
| | 880 | 7134 | 0.82 | 0.13 | 0.85 | 0.04 | 1.00 | -2.29 | 0.57 | 0.79 | 0.85 | 0.89 | 0.95 | |
| $PM_{10}$ | - | 18782 | 11.32 | 9.93 | 8.46 | 0.10 | 153.62 | 2.60 | 1.36 | 4.84 | 8.46 | 15.54 | 29.51 | |

a)

[Figure]

b)

[Figure]

**Figure S1.** Frequency distribution, average and standard deviation of SAE, AAE, $PM_{10}$ and $\%PM_{1-10}$ in $PM_{10}$ parameters for the three atmospheric scenarios (SDE, REG, AA) displayed in the Ångström matrix at (a) MSY and (b) MSA.

[Figure]

40    **Figure S21.** Ångström matrix (AAE vs. SAE weighted by % dust in PM$_{10}$) during Saharan dust events at MSY (daily base).

[Figure]

a)

NOAA HYSPLIT MODEL
Backward trajectories ending at 1200 UTC 01 Nov 13
GFSG Meteorological Data

b)

NOAA HYSPLIT MODEL
Backward trajectories ending at 1200 UTC 28 Oct 13
GFSG Meteorological Data

c)

BSC-DREAM8b v2.0 Dust Low Level Conc. ($\mu g/m^3$ )
00h forecast for 12UTC 28 Oct 2013
http://www.bsc.es/projects/earthscience/BSC-DREAM/

[Figure]

**Figure S32.** Backward trajectories corresponding to (a) the Saharan dust event (28/10/2013) and (b) the regional episode (01/11/2013) atmospheric scenarios at MSY. Dust surface concentration at MSY from the Dream model corresponding to (c) the Saharan dust event (28/10/2013) and (d) the regional episode (01/11/2013).

[Figure]

**Figure S43.** Summer and winter daily cycles of: (a) AAE at MSY and MSA, (b) measured OM and simulated OM as the sum of OMff and OMbb contributions at MSY, measured BC and simulated BC as the sum of BCff and BCbb contributions at (c) MSY and (d) MSA. Averages were calculated from hourly base.